# Interpretable Distribution Features
# with Maximum Testing Power

**Wittawat Jitkrittum,  Zoltán Szabó,  Kacper Chwialkowski,  Arthur Gretton**
`wittawatj@gmail.com`
`zoltan.szabo.m@gmail.com`
`kacper.chwialkowski@gmail.com`
`arthur.gretton@gmail.com`

Gatsby Unit, University College London

## Abstract

Two semimetrics on probability distributions are proposed, given as the sum of differences of expectations of analytic functions evaluated at spatial or frequency locations (i.e, features). The features are chosen so as to maximize the distinguishability of the distributions, by optimizing a lower bound on test power for a statistical test using these features. The result is a parsimonious and interpretable indication of how and where two distributions differ locally. We show that the empirical estimate of the test power criterion converges with increasing sample size, ensuring the quality of the returned features. In real-world benchmarks on high-dimensional text and image data, linear-time tests using the proposed semimetrics achieve comparable performance to the state-of-the-art quadratic-time maximum mean discrepancy test, while returning human-interpretable features that explain the test results.

## 1  Introduction

We address the problem of discovering features of distinct probability distributions, with which they can most easily be distinguished. The distributions may be in high dimensions, can differ in non-trivial ways (i.e., not simply in their means), and are observed only through i.i.d. samples. One application for such divergence measures is to model criticism, where samples from a trained model are compared with a validation sample: in the univariate case, through the KL divergence (Cinzia Carota and Polson, 1996), or in the multivariate case, by use of the maximum mean discrepancy (MMD) (Lloyd and Ghahramani, 2015). An alternative, interpretable analysis of a multivariate difference in distributions may be obtained by projecting onto a discriminative direction, such that the Wasserstein distance on this projection is maximized (Mueller and Jaakkola, 2015). Note that both recent works require low dimensionality, either explicitly (in the case of Lloyd and Gharamani, the function becomes difficult to plot in more than two dimensions), or implicitly in the case of Mueller and Jaakkola, in that a large difference in distributions must occur in projection along a particular one-dimensional axis. Distances between distributions in high dimensions may be more subtle, however, and it is of interest to find interpretable, distinguishing features of these distributions.

In the present paper, we take a hypothesis testing approach to discovering features which best distinguish two multivariate probability measures $P$ and $Q$, as observed by samples $\mathsf{X} := \{\mathbf{x}_i\}_{i=1}^n$ drawn independently and identically (i.i.d.) from $P$, and $\mathsf{Y} := \{\mathbf{y}_i\}_{i=1}^n \subset \mathbb{R}^d$ from $Q$. Non-parametric two-sample tests based on RKHS distances (Eric et al., 2008; Fromont et al., 2012; Gretton et al., 2012a) or energy distances (Székely and Rizzo, 2004; Baringhaus and Franz, 2004) have as their test statistic an integral probability metric, the Maximum Mean Discrepancy (Gretton et al., 2012a; Sejdinovic et al., 2013). For this metric, a smooth witness function is computed, such that the amplitude is largest where the probability mass differs most (e.g. Gretton et al., 2012a,

Figure 1). Lloyd and Ghahramani (2015) used this witness function to compare the model output of the Automated Statistician (Lloyd et al., 2014) with a reference sample, yielding a visual indication of where the model fails. In high dimensions, however, the witness function cannot be plotted, and is less helpful. Furthermore, the witness function does not give an easily interpretable result for distributions with local differences in their characteristic functions. A more subtle shortcoming is that it does not provide a direct indication of the distribution features which, when compared, would maximize test power - rather, it is the witness function *norm*, and (broadly speaking) its *variance* under the null, that determine test power.

Our approach builds on the analytic representations of probability distributions of Chwialkowski et al. (2015), where differences in expectations of analytic functions at particular spatial or frequency locations are used to construct a two-sample test statistic, which can be computed in linear time. Despite the differences in these analytic functions being evaluated at random locations, the analytic tests have greater power than linear time tests based on subsampled estimates of the MMD (Gretton et al., 2012b; Zaremba et al., 2013). Our first theoretical contribution, in Sec. 3, is to derive a lower bound on the test power, which can be maximized over the choice of test locations. We propose two novel tests, both of which significantly outperform the random feature choice of Chwialkowski et al.. The (ME) test evaluates the difference of mean embeddings at locations chosen to maximize the test power lower bound (i.e., spatial features); unlike the maxima of the MMD witness function, these features are directly chosen to maximize the distinguishability of the distributions, and take variance into account. The Smooth Characteristic Function (SCF) test uses as its statistic the difference of the two smoothed empirical characteristic functions, evaluated at points in the frequency domain so as to maximize the same criterion (i.e., frequency features). Optimization of the mean embedding kernels/frequency smoothing functions themselves is achieved on a held-out data set with the same consistent objective.

As our second theoretical contribution in Sec. 3, we prove that the empirical estimate of the test power criterion asymptotically converges to its population quantity uniformly over the class of Gaussian kernels. Two important consequences follow: first, in testing, we obtain a more powerful test with fewer features. Second, we obtain a parsimonious and interpretable set of features that best distinguish the probability distributions. In Sec. 4, we provide experiments demonstrating that the proposed linear-time tests greatly outperform all previous linear time tests, and achieve performance that compares to or exceeds the more expensive quadratic-time MMD test (Gretton et al., 2012a). Moreover, the new tests discover features of text data (NIPS proceedings) and image data (distinct facial expressions) which have a clear human interpretation, thus validating our feature elicitation procedure in these challenging high-dimensional testing scenarios.

## 2  ME and SCF tests

In this section, we review the ME and SCF tests (Chwialkowski et al., 2015) for two-sample testing. In Sec. 3, we will extend these approaches to learn features that optimize the power of these tests. Given two samples $\mathsf{X} := \{\mathbf{x}_i\}_{i=1}^n, \mathsf{Y} := \{\mathbf{y}_i\}_{i=1}^n \subset \mathbb{R}^d$ independently and identically distributed (i.i.d.) according to $P$ and $Q$, respectively, the goal of a two-sample test is to decide whether $P$ is different from $Q$ on the basis of the samples. The task is formulated as a statistical hypothesis test proposing a null hypothesis $H_0 : P = Q$ (samples are drawn from the same distribution) against an alternative hypothesis $H_1 : P \neq Q$ (the sample generating distributions are different). A test calculates a test statistic $\hat{\lambda}_n$ from $\mathsf{X}$ and $\mathsf{Y}$, and rejects $H_0$ if $\hat{\lambda}_n$ exceeds a predetermined test threshold (critical value). The threshold is given by the $(1 - \alpha)$-quantile of the (asymptotic) distribution of $\hat{\lambda}_n$ under $H_0$ i.e., the null distribution, and $\alpha$ is the significance level of the test.

**ME test**  The ME test uses as its test statistic $\hat{\lambda}_n$, a form of Hotelling's T-squared statistic, defined as $\hat{\lambda}_n := n\bar{\mathbf{z}}_n^\top \mathbf{S}_n^{-1} \bar{\mathbf{z}}_n$, where $\bar{\mathbf{z}}_n := \frac{1}{n}\sum_{i=1}^n \mathbf{z}_i$, $\mathbf{S}_n := \frac{1}{n-1}\sum_{i=1}^n (\mathbf{z}_i - \bar{\mathbf{z}}_n)(\mathbf{z}_i - \bar{\mathbf{z}}_n)^\top$, and $\mathbf{z}_i := (k(\mathbf{x}_i, \mathbf{v}_j) - k(\mathbf{y}_i, \mathbf{v}_j))_{j=1}^J \in \mathbb{R}^J$. The statistic depends on a positive definite kernel $k : \mathcal{X} \times \mathcal{X} \to \mathbb{R}$ (with $\mathcal{X} \subseteq \mathbb{R}^d$), and a set of $J$ test locations $\mathcal{V} = \{\mathbf{v}_j\}_{j=1}^J \subset \mathbb{R}^d$. Under $H_0$, asymptotically $\hat{\lambda}_n$ follows $\chi^2(J)$, a chi-squared distribution with $J$ degrees of freedom. The ME test rejects $H_0$ if $\hat{\lambda}_n > T_\alpha$, where the test threshold $T_\alpha$ is given by the $(1 - \alpha)$-quantile of the asymptotic null distribution $\chi^2(J)$. Although the distribution of $\hat{\lambda}_n$ under $H_1$ was not derived, Chwialkowski et al. (2015) showed that if $k$ is analytic, integrable and characteristic (in the sense of Sriperumbudur et al. (2011)), under $H_1$, $\hat{\lambda}_n$ can be arbitrarily large as $n \to \infty$, allowing the test to correctly reject $H_0$.

One can intuitively think of the ME test statistic as a squared normalized (by the inverse covariance $\mathbf{S}_n^{-1}$) $L^2(\mathcal{X}, V_J)$ distance of the mean embeddings (Smola et al., 2007) of the empirical measures $P_n := \frac{1}{n}\sum_{i=1}^n \delta_{\mathbf{x}_i}$, and $Q_n := \frac{1}{n}\sum_{i=1}^n \delta_{\mathbf{y}_i}$ where $V_J := \frac{1}{J}\sum_{j=1}^J \delta_{\mathbf{v}_j}$, and $\delta_{\mathbf{x}}$ is the Dirac measure concentrated at $\mathbf{x}$. The unnormalized counterpart (i.e., without $\mathbf{S}_n^{-1}$) was shown by Chwialkowski et al. (2015) to be a metric on the space of probability measures for any $\mathcal{V}$. Both variants behave similarly for two-sample testing, with the normalized version being a semimetric having a more computationally tractable null distribution, i.e., $\chi^2(J)$.

**SCF test** The SCF uses the test statistic which has the same form as the ME test statistic with a modified $\mathbf{z}_i := [\hat{l}(\mathbf{x}_i)\sin(\mathbf{x}_i^\top \mathbf{v}_j) - \hat{l}(\mathbf{y}_i)\sin(\mathbf{y}_i^\top \mathbf{v}_j), \hat{l}(\mathbf{x}_i)\cos(\mathbf{x}_i^\top \mathbf{v}_j) - \hat{l}(\mathbf{y}_i)\cos(\mathbf{y}_i^\top \mathbf{v}_j)]_{j=1}^J \in \mathbb{R}^{2J}$, where $\hat{l}(\mathbf{x}) = \int_{\mathbb{R}^d} \exp(-i\mathbf{u}^\top \mathbf{x})l(\mathbf{u})\,\mathrm{d}\mathbf{u}$ is the Fourier transform of $l(\mathbf{x})$, and $l : \mathbb{R}^d \to \mathbb{R}$ is an analytic translation-invariant kernel i.e., $l(\mathbf{x} - \mathbf{y})$ defines a positive definite kernel for $\mathbf{x}$ and $\mathbf{y}$. In contrast to the ME test defining the statistic in terms of spatial locations, the locations $\mathcal{V} = \{\mathbf{v}_j\}_{j=1}^J \subset \mathbb{R}^d$ in the SCF test are in the frequency domain. As a brief description, let $\varphi_P(\mathbf{w}) := \mathbb{E}_{\mathbf{x}\sim P}\exp(i\mathbf{w}^\top\mathbf{x})$ be the characteristic function of $P$. Define a smooth characteristic function as $\phi_P(\mathbf{v}) = \int_{\mathbb{R}^d}\varphi_P(\mathbf{w})l(\mathbf{v} - \mathbf{w})\,\mathrm{d}\mathbf{w}$ (Chwialkowski et al., 2015, Definition 2). Then, similar to the ME test, the statistic defined by the SCF test can be seen as a normalized (by $\mathbf{S}_n^{-1}$) version of $L^2(\mathcal{X}, V_J)$ distance of empirical $\phi_P(\mathbf{v})$ and $\phi_Q(\mathbf{v})$. The SCF test statistic has asymptotic distribution $\chi^2(2J)$ under $H_0$. We will use $J'$ to refer to the degrees of freedom of the chi-squared distribution i.e., $J' = J$ for the ME test, and $J' = 2J$ for the SCF test.

In this work, we modify the statistic with a regularization parameter $\gamma_n > 0$, giving $\hat{\lambda}_n := n\bar{\mathbf{z}}_n^\top (\mathbf{S}_n + \gamma_n I)^{-1}\bar{\mathbf{z}}_n$, for stability of the matrix inverse. Using multivariate Slutsky's theorem, under $H_0$, $\hat{\lambda}_n$ still asymptotically follows $\chi^2(J')$ provided that $\gamma_n \to 0$ as $n \to \infty$.

## 3 Lower bound on test power, consistency of empirical power statistic

This section contains our main results. We propose to optimize the test locations $\mathcal{V}$ and kernel parameters (jointly referred to as $\theta$) by maximizing a lower bound on the test power in Proposition 1. This criterion offers a simple objective function for fast parameter tuning. The bound may be of independent interest in other Hotelling's T-squared statistics, since apart from the Gaussian case (e.g. Bilodeau and Brenner, 2008, Ch. 8), the characterization of such statistics under the alternative distribution is challenging. The optimization procedure is given in Sec. 4. We use $\mathbb{E}_{\mathbf{xy}}$ as a shorthand for $\mathbb{E}_{\mathbf{x}\sim P}\mathbb{E}_{\mathbf{y}\sim Q}$ and let $\|\cdot\|_F$ be the Frobenius norm.

**Proposition 1** (Lower bound on ME test power). *Let $\mathcal{K}$ be a uniformly bounded (i.e., $\exists B < \infty$ such that $\sup_{k\in\mathcal{K}}\sup_{(\mathbf{x},\mathbf{y})\in\mathcal{X}^2}|k(\mathbf{x},\mathbf{y})| \leq B$) family of $k : \mathcal{X}\times\mathcal{X} \to \mathbb{R}$ measurable kernels. Let $\mathbb{V}$ be a collection in which each element is a set of $J$ test locations. Assume that $\tilde{c} := \sup_{\mathcal{V}\in\mathbb{V}, k\in\mathcal{K}}\|\Sigma^{-1}\|_F < \infty$. Then, the test power $\mathbb{P}\left(\hat{\lambda}_n \geq T_\alpha\right)$ of the ME test satisfies $\mathbb{P}\left(\hat{\lambda}_n \geq T_\alpha\right) \geq L(\lambda_n)$ where*

$$L(\lambda_n) := 1 - 2e^{-\xi_1(\lambda_n - T_\alpha)^2/n} - 2e^{-\frac{[\gamma_n(\lambda_n - T_\alpha)(n-1) - \xi_2 n]^2}{\xi_3 n(2n-1)^2}} - 2e^{-[(\lambda_n - T_\alpha)/3 - \bar{c}_3 n\gamma_n]^2\gamma_n^2/\xi_4},$$

*and $\bar{c}_3, \xi_1, \ldots \xi_4$ are positive constants depending on only $B, J$ and $\tilde{c}$. The parameter $\lambda_n := n\boldsymbol{\mu}^\top\Sigma^{-1}\boldsymbol{\mu}$ is the population counterpart of $\hat{\lambda}_n := n\bar{\mathbf{z}}_n^\top(\mathbf{S}_n + \gamma_n I)^{-1}\bar{\mathbf{z}}_n$ where $\boldsymbol{\mu} = \mathbb{E}_{\mathbf{xy}}[\mathbf{z}_1]$ and $\Sigma = \mathbb{E}_{\mathbf{xy}}[(\mathbf{z}_1 - \boldsymbol{\mu})(\mathbf{z}_1 - \boldsymbol{\mu})^\top]$. For large $n$, $L(\lambda_n)$ is increasing in $\lambda_n$.*

*Proof (sketch).* The idea is to construct a bound for $|\hat{\lambda}_n - \lambda_n|$ which involves bounding $\|\bar{\mathbf{z}}_n - \boldsymbol{\mu}\|_2$ and $\|\mathbf{S}_n - \Sigma\|_F$ separately using Hoeffding's inequality. The result follows after a reparameterization of the bound on $\mathbb{P}(|\hat{\lambda}_n - \lambda_n| \geq t)$ to have $\mathbb{P}\left(\hat{\lambda}_n \geq T_\alpha\right)$. See Sec. F for details. $\qquad\square$

Proposition 1 suggests that for large $n$ it is sufficient to maximize $\lambda_n$ to maximize a lower bound on the ME test power. The same conclusion holds for the SCF test (result omitted due to space constraints). Assume that $k$ is characteristic (Sriperumbudur et al., 2011). It can be shown that $\lambda_n = 0$ if and only if $P = Q$ i.e., $\lambda_n$ is a semimetric for $P$ and $Q$. In this sense, one can see $\lambda_n$ as encoding the ease of rejecting $H_0$. The higher $\lambda_n$, the easier for the test to correctly reject $H_0$ when $H_1$ holds. This observation justifies the use of $\lambda_n$ as a maximization objective for parameter tuning.

**Contributions** The statistic $\hat{\lambda}_n$ for both ME and SCF tests depends on a set of test locations $\mathcal{V}$ and a kernel parameter $\sigma$. We propose to set $\theta := \{\mathcal{V}, \sigma\} = \arg\max_\theta \lambda_n = \arg\max_\theta \boldsymbol{\mu}^\top \boldsymbol{\Sigma}^{-1} \boldsymbol{\mu}$. The optimization of $\theta$ brings two benefits: first, it significantly increases the probability of rejecting $H_0$ when $H_1$ holds; second, the learned test locations act as discriminative features allowing an interpretation of how the two distributions differ. We note that optimizing parameters by maximizing a test power proxy (Gretton et al., 2012b) is valid under both $H_0$ and $H_1$ as long as the data used for parameter tuning and for testing are disjoint. If $H_0$ holds, then $\theta = \arg\max 0$ is arbitrary. Since the test statistic asymptotically follows $\chi^2(J')$ for any $\theta$, the optimization does not change the null distribution. Also, the rejection threshold $T_\alpha$ depends on only $J'$ and is independent of $\theta$.

To avoid creating a dependency between $\theta$ and the data used for testing (which would affect the null distribution), we split the data into two disjoint sets. Let $\mathsf{D} := (\mathsf{X}, \mathsf{Y})$ and $\mathsf{D}^{tr}, \mathsf{D}^{te} \subset \mathsf{D}$ such that $\mathsf{D}^{tr} \cap \mathsf{D}^{te} = \emptyset$ and $\mathsf{D}^{tr} \cup \mathsf{D}^{te} = \mathsf{D}$. In practice, since $\boldsymbol{\mu}$ and $\boldsymbol{\Sigma}$ are unknown, we use $\hat{\lambda}^{tr}_{n/2}$ in place of $\lambda_n$, where $\hat{\lambda}^{tr}_{n/2}$ is the test statistic computed on the training set $\mathsf{D}^{tr}$. For simplicity, we assume that each of $\mathsf{D}^{tr}$ and $\mathsf{D}^{te}$ has half of the samples in $\mathsf{D}$. We perform an optimization of $\theta$ with gradient ascent algorithm on $\hat{\lambda}^{tr}_{n/2}(\theta)$. The actual two-sample test is performed using the test statistic $\hat{\lambda}^{te}_{n/2}(\theta)$ computed on $\mathsf{D}^{te}$. The full procedure from tuning the parameters to the actual two-sample test is summarized in Sec. A.

Since we use an empirical estimate $\hat{\lambda}^{tr}_{n/2}$ in place of $\lambda_n$ for parameter optimization, we give a finite-sample bound in Theorem 2 guaranteeing the convergence of $\bar{\mathbf{z}}_n^\top (\mathbf{S}_n + \gamma_n I)^{-1} \bar{\mathbf{z}}_n$ to $\boldsymbol{\mu}^\top \boldsymbol{\Sigma}^{-1} \boldsymbol{\mu}$ as $n$ increases, uniformly over all kernels $k \in \mathcal{K}$ (a family of uniformly bounded kernels) and all test locations in an appropriate class. Kernel classes satisfying conditions of Theorem 2 include the widely used isotropic Gaussian kernel class $\mathcal{K}_g = \left\{ k_\sigma : (\mathbf{x}, \mathbf{y}) \mapsto \exp\left(-(2\sigma^2)^{-1} \|\mathbf{x} - \mathbf{y}\|^2\right) \mid \sigma > 0 \right\}$, and the more general full Gaussian kernel class $\mathcal{K}_{\text{full}} = \{ k : (\mathbf{x}, \mathbf{y}) \mapsto \exp\left(-(\mathbf{x} - \mathbf{y})^\top \mathbf{A}(\mathbf{x} - \mathbf{y})\right) \mid \mathbf{A}$ is positive definite$\}$ (see Lemma 5 and Lemma 6).

**Theorem 2** (Consistency of $\hat{\lambda}_n$ in the ME test). *Let $\mathcal{X} \subseteq \mathbb{R}^d$ be a measurable set, and $\mathbb{V}$ be a collection in which each element is a set of $J$ test locations. All suprema over $\mathcal{V}$ and $k$ are to be understood as $\sup_{\mathcal{V} \in \mathbb{V}}$ and $\sup_{k \in \mathcal{K}}$ respectively. For a class of kernels $\mathcal{K}$ on $\mathcal{X} \subseteq \mathbb{R}^d$, define*

$$\mathcal{F}_1 := \{\mathbf{x} \mapsto k(\mathbf{x}, \mathbf{v}) \mid k \in \mathcal{K}, \mathbf{v} \in \mathcal{X}\}, \quad \mathcal{F}_2 := \{\mathbf{x} \mapsto k(\mathbf{x}, \mathbf{v}) k(\mathbf{x}, \mathbf{v}') \mid k \in \mathcal{K}, \mathbf{v}, \mathbf{v}' \in \mathcal{X}\}, \quad (1)$$

$$\mathcal{F}_3 := \{(\mathbf{x}, \mathbf{y}) \mapsto k(\mathbf{x}, \mathbf{v}) k(\mathbf{y}, \mathbf{v}') \mid k \in \mathcal{K}, \mathbf{v}, \mathbf{v}' \in \mathcal{X}\}. \quad (2)$$

*Assume that (1) $\mathcal{K}$ is a uniformly bounded (by $B$) family of $k : \mathcal{X} \times \mathcal{X} \to \mathbb{R}$ measurable kernels, (2) $\tilde{c} := \sup_{\mathcal{V}, k} \|\boldsymbol{\Sigma}^{-1}\|_F < \infty$, and (3) $\mathcal{F}_i = \{f_{\theta_i} \mid \theta_i \in \Theta_i\}$ is VC-subgraph with VC-index $VC(\mathcal{F}_i)$, and $\theta \mapsto f_{\theta_i}(m)$ is continuous $(\forall m, i = 1, 2, 3)$. Let $\bar{c}_1 := 4B^2 J \sqrt{J} \tilde{c}, \bar{c}_2 := 4B\sqrt{J}\tilde{c}$, and $\bar{c}_3 := 4B^2 J \tilde{c}^2$. Let $C_i$-s $(i = 1, 2, 3)$ be the universal constants associated to $\mathcal{F}_i$-s according to Theorem 2.6.7 in van der Vaart and Wellner (2000). Then for any $\delta \in (0, 1)$ with probability at least $1 - \delta$,*

$$\sup_{\mathcal{V}, k} \left| \bar{\mathbf{z}}_n^\top (\mathbf{S}_n + \gamma_n I)^{-1} \bar{\mathbf{z}}_n - \boldsymbol{\mu}^\top \boldsymbol{\Sigma}^{-1} \boldsymbol{\mu} \right|$$

$$\leq 2 T_{\mathcal{F}_1} \left( \frac{2}{\gamma_n} \bar{c}_1 B J \frac{2n-1}{n-1} + \bar{c}_2 \sqrt{J} \right) + \frac{2}{\gamma_n} \bar{c}_1 J (T_{\mathcal{F}_2} + T_{\mathcal{F}_3}) + \frac{8}{\gamma_n} \frac{\bar{c}_1 B^2 J}{n-1} + \bar{c}_3 \gamma_n, \textit{where}$$

$$T_{\mathcal{F}_j} = \frac{16\sqrt{2} B^{\zeta_j}}{\sqrt{n}} \left( 2\sqrt{\log\left[C_j \times VC(\mathcal{F}_j)(16e)^{VC(\mathcal{F}_j)}\right]} + \frac{\sqrt{2\pi[VC(\mathcal{F}_j) - 1]}}{2} \right) + B^{\zeta_j} \sqrt{\frac{2\log(5/\delta)}{n}},$$

*for $j = 1, 2, 3$ and $\zeta_1 = 1, \zeta_2 = \zeta_3 = 2$.*

*Proof (sketch).* The idea is to lower bound the difference with an expression involving $\sup_{\mathcal{V}, k} \|\bar{\mathbf{z}}_n - \boldsymbol{\mu}\|_2$ and $\sup_{\mathcal{V}, k} \|\mathbf{S}_n - \boldsymbol{\Sigma}\|_F$. These two quantities can be seen as suprema of empirical processes, and can be bounded by Rademacher complexities of their respective function classes (i.e., $\mathcal{F}_1, \mathcal{F}_2$, and $\mathcal{F}_3$). Finally, the Rademacher complexities can be upper bounded using Dudley entropy bound and VC subgraph properties of the function classes. Proof details are given in Sec. D. □

Theorem 2 implies that if we set $\gamma_n = \mathcal{O}(n^{-1/4})$, then we have $\sup_{\mathcal{V}, k} \left| \bar{\mathbf{z}}_n^\top (\mathbf{S}_n + \gamma_n I)^{-1} \bar{\mathbf{z}}_n - \boldsymbol{\mu}^\top \boldsymbol{\Sigma}^{-1} \boldsymbol{\mu} \right| = \mathcal{O}_p(n^{-1/4})$ as the rate of convergence. Both

Proposition 1 and Theorem 2 require $\tilde{c} := \sup_{\mathcal{V}\in\mathbb{V}, k\in\mathcal{K}} \|\mathbf{\Sigma}^{-1}\|_F < \infty$ as a precondition. To guarantee that $\tilde{c} < \infty$, a concrete construction of $\mathcal{K}$ is the isotropic Gaussian kernel class $\mathcal{K}_g$, where $\sigma$ is constrained to be in a compact set. Also, consider $\mathbb{V} := \{\mathcal{V} \mid$ any two locations are at least $\epsilon$ distance apart, and all test locations have their norms bounded by $\zeta\}$ for some $\epsilon, \zeta > 0$. Then, for any non-degenerate $P, Q$, we have $\tilde{c} < \infty$ since $(\sigma, \mathcal{V}) \mapsto \lambda_n$ is continuous, and thus attains its supremum over compact sets $\mathcal{K}$ and $\mathbb{V}$.

## 4 Experiments

In this section, we demonstrate the effectiveness of the proposed methods on both toy and real problems. We consider the isotropic Gaussian kernel class $\mathcal{K}_g$ in all kernel-based tests. We study seven two-sample test algorithms. For the SCF test, we set $\hat{l}(\mathbf{x}) = k(\mathbf{x}, \mathbf{0})$. Denote by ME-full and SCF-full the ME and SCF tests whose test locations and the Gaussian width $\sigma$ are fully optimized using gradient ascent on a separate training sample ($\mathrm{D}^{tr}$) of the same size as the test set ($\mathrm{D}^{te}$). ME-grid and SCF-grid are as in Chwialkowski et al. (2015) where only the Gaussian width is optimized by a grid search,[1] and the test locations are randomly drawn from a multivariate normal distribution. MMD-quad (quadratic-time) and MMD-lin (linear-time) re-

Table 1: Four toy problems. $H_0$ holds only in SG.

| Data | $P$ | $Q$ |
|---|---|---|
| SG | $\mathcal{N}(\mathbf{0}_d, I_d)$ | $\mathcal{N}(\mathbf{0}_d, I_d)$ |
| GMD | $\mathcal{N}(\mathbf{0}_d, I_d)$ | $\mathcal{N}((1, 0, \ldots, 0)^\top, I_d)$ |
| GVD | $\mathcal{N}(\mathbf{0}_d, I_d)$ | $\mathcal{N}(\mathbf{0}_d, \mathrm{diag}(2, 1, \ldots, 1))$ |
| Blobs | Gaussian mixtures in $\mathbb{R}^2$ as studied in Chwialkowski et al. (2015); Gretton et al. (2012b). | |

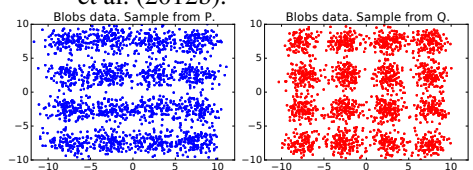

fer to the nonparametric tests based on maximum mean discrepancy of Gretton et al. (2012a), where to ensure a fair comparison, the Gaussian kernel width is also chosen so as to maximize a criterion for the test power on training data, following the same principle as (Gretton et al., 2012b). For MMD-quad, since its null distribution is given by an infinite sum of weighted chi-squared variables (no closed-form quantiles), in each trial we randomly permute the two samples 400 times to approximate the null distribution. Finally, $T^2$ is the standard two-sample Hotelling's T-squared test, which serves as a baseline with Gaussian assumptions on $P$ and $Q$.

In all the following experiments, each problem is repeated for 500 trials. For toy problems, new samples are generated from the specified $P, Q$ distributions in each trial. For real problems, samples are partitioned randomly into training and test sets in each trial. In all of the simulations, we report an empirical estimate of $\mathbb{P}(\hat{\lambda}_{n/2}^{te} \geq T_\alpha)$ which is the proportion of the number of times the statistic $\hat{\lambda}_{n/2}^{te}$ is above $T_\alpha$. This quantity is an estimate of type-I error under $H_0$, and corresponds to test power when $H_1$ is true. We set $\alpha = 0.01$ in all the experiments. All the code and preprocessed data are available at https://github.com/wittawatj/interpretable-test.

**Optimization** The parameter tuning objective $\hat{\lambda}_{n/2}^{tr}(\theta)$ is a function of $\theta$ consisting of one real-valued $\sigma$ and $J$ test locations each of $d$ dimensions. The parameters $\theta$ can thus be regarded as a $Jd + 1$ Euclidean vector. We take the derivative of $\hat{\lambda}_{n/2}^{tr}(\theta)$ with respect to $\theta$, and use gradient ascent to maximize it. $J$ is pre-specified and fixed. For the ME test, we initialize the test locations with realizations from two multivariate normal distributions fitted to samples from $P$ and $Q$; this ensures that the initial locations are well supported by the data. For the SCF test, initialization using the standard normal distribution is found to be sufficient. The parameter $\gamma_n$ is not optimized; we set the regularization parameter $\gamma_n$ to be as small as possible while being large enough to ensure that $(\mathbf{S}_n + \gamma_n I)^{-1}$ can be stably computed. We emphasize that both the optimization and testing are linear in $n$. The testing cost $\mathcal{O}(J^3 + J^2 n + dJn)$ and the optimization costs $\mathcal{O}(J^3 + dJ^2 n)$ per gradient ascent iteration. Runtimes of all methods are reported in Sec. C in the appendix.

**1. Informative features: simple demonstration** We begin with a demonstration that the proxy $\hat{\lambda}_{n/2}^{tr}(\theta)$ for the test power is informative for revealing the difference of the two samples in the ME

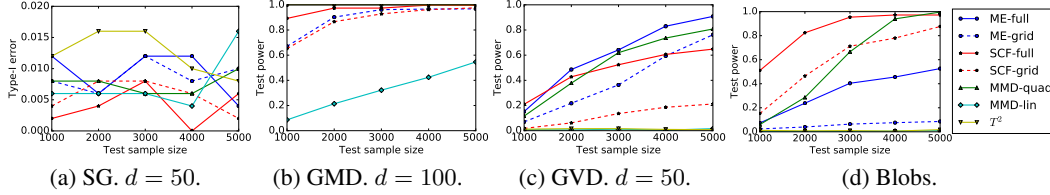

| | | | |
|---|---|---|---|
| (a) SG. $d = 50$. | (b) GMD. $d = 100$. | (c) GVD. $d = 50$. | (d) Blobs. |

Figure 2: Plots of type-I error/test power against the test sample size $n^{te}$ in the four toy problems.

test. We consider the Gaussian Mean Difference (GMD) problem (see Table 1), where both $P$ and $Q$ are two-dimensional normal distributions with the difference in means. We use $J = 2$ test locations $\mathbf{v}_1$ and $\mathbf{v}_2$, where $\mathbf{v}_1$ is fixed to the location indicated by the black triangle in Fig. 1. The contour plot shows $\mathbf{v}_2 \mapsto \hat{\lambda}^{tr}_{n/2}(\mathbf{v}_1, \mathbf{v}_2)$.

Fig. 1 (top) suggests that $\hat{\lambda}^{tr}_{n/2}$ is maximized when $\mathbf{v}_2$ is placed in either of the two regions that captures the difference of the two samples i.e., the region in which the probability masses of $P$ and $Q$ have less overlap. Fig. 1 (bottom), we consider placing $\mathbf{v}_1$ in one of the two key regions. In this case, the contour plot shows that $\mathbf{v}_2$ should be placed in the other region to maximize $\hat{\lambda}^{tr}_{n/2}$, implying that placing multiple test locations in the same neighborhood will not increase the discriminability. The two modes on the left and right suggest two ways to place the test location in a region that reveals the difference. The non-convexity of the $\hat{\lambda}^{tr}_{n/2}$ is an indication of many informative ways to detect differences of $P$ and $Q$, rather than a drawback. A convex objective would not capture this multimodality.

**2. Test power vs. sample size** $n$ We now demonstrate the rate of increase of test power with sample size. When the null hypothesis holds, the type-I error stays at the specified level $\alpha$. We consider the following four toy problems: Same Gaussian (SG), Gaussian mean difference (GMD), Gaussian variance difference (GVD), and Blobs. The specifications of $P$ and $Q$ are summarized in Table. 1. In the Blobs problem, $P$ and $Q$ are defined as a mixture of Gaussian distributions arranged on a $4 \times 4$ grid in $\mathbb{R}^2$. This problem is challenging as the difference of $P$ and $Q$ is encoded at a much smaller length scale compared to the global structure (Gretton et al., 2012b). Specifically, the eigenvalue ratio for the covariance of each Gaussian distribution is 2.0 in $P$, and 1.0 in $Q$. We set $J = 5$ in this experiment.

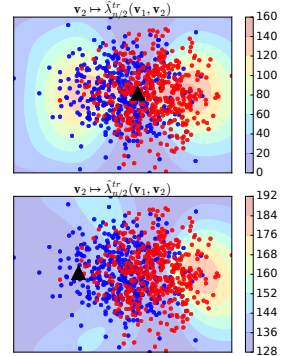

Figure 1: A contour plot of $\hat{\lambda}^{tr}_{n/2}$ as a function of $\mathbf{v}_2$ when $J = 2$ and $\mathbf{v}_1$ is fixed (black triangle). The objective $\hat{\lambda}^{tr}_{n/2}$ is high in the regions that reveal the difference of the two samples.

The results are shown in Fig. 2 where type-I error (for SG problem), and test power (for GMD, GVD and Blobs problems) are plotted against test sample size. A number of observations are worth noting. In the SG problem, we see that the type-I error roughly stays at the specified level: the rate of rejection of $H_0$ when it is true is roughly at the specified level $\alpha = 0.01$.

GMD with 100 dimensions turns out to be an easy problem for all the tests except MMD-lin. In the GVD and Blobs cases, ME-full and SCF-full achieve substantially higher test power than ME-grid and SCF-grid, respectively, suggesting a clear advantage from optimizing the test locations. Remarkably, ME-full consistently outperforms the quadratic-time MMD across all test sample sizes in the GVD case. When the difference of $P$ and $Q$ is subtle as in the Blobs problem, ME-grid, which uses randomly drawn test locations, can perform poorly (see Fig. 2d) since it is unlikely that randomly drawn locations will be placed in the key regions that reveal the difference. In this case, optimization of the test locations can considerably boost the test power (see ME-full in Fig. 2d). Note also that SCF variants perform significantly better than ME variants on the Blobs problem, as the difference in $P$ and $Q$ is localized in the frequency domain; ME-full and ME-grid would require many more test locations in the spatial domain to match the test powers of the SCF variants. For the same reason, SCF-full does much better than the quadratic-time MMD across most sample sizes, as the latter represents a weighted distance between characteristic functions integrated across the entire frequency domain (Sriperumbudur et al., 2010, Corollary 4).

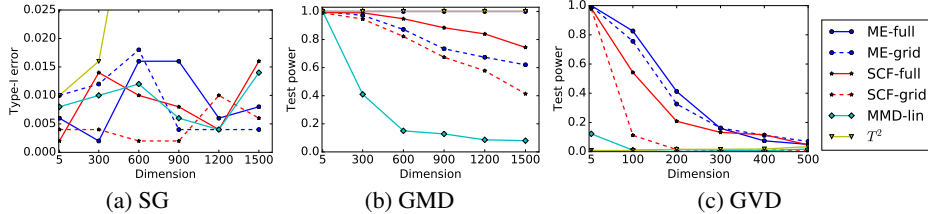

(a) SG        (b) GMD        (c) GVD

Figure 3: Plots of type-I error/test power against the dimensions $d$ in the four toy problems in Table 1.

Table 2: Type-I errors and powers of various tests in the problem of distinguishing NIPS papers from two categories. $\alpha = 0.01$. $J = 1$. $n_{te}$ denotes the test sample size of each of the two samples.

| Problem | $n^{te}$ | ME-full | ME-grid | SCF-full | SCF-grid | MMD-quad | MMD-lin |
|---|---|---|---|---|---|---|---|
| Bayes-Bayes | 215 | .012 | .018 | .012 | .004 | .022 | .008 |
| Bayes-Deep | 216 | .954 | .034 | .688 | .180 | .906 | .262 |
| Bayes-Learn | 138 | .990 | .774 | .836 | .534 | 1.00 | .238 |
| Bayes-Neuro | 394 | 1.00 | .300 | .828 | .500 | .952 | .972 |
| Learn-Deep | 149 | .956 | .052 | .656 | .138 | .876 | .500 |
| Learn-Neuro | 146 | .960 | .572 | .590 | .360 | 1.00 | .538 |

**3. Test power vs. dimension** $d$    We next investigate how the dimension $(d)$ of the problem can affect type-I errors and test powers of ME and SCF tests. We consider the same artificial problems: SG, GMD and GVD. This time, we fix the test sample size to 10000, set $J = 5$, and vary the dimension. The results are shown in Fig. 3. Due to the large dimensions and sample size, it is computationally infeasible to run MMD-quad.

We observe that all the tests except the T-test can maintain type-I error at roughly the specified significance level $\alpha = 0.01$ as dimension increases. The type-I performance of the T-test is incorrect at large $d$ because of the difficulty in accurately estimating the covariance matrix in high dimensions. It is interesting to note the high performance of ME-full in the GMD problem in Fig. 3b. ME-full achieves the maximum test power of 1.0 throughout and matches the power T-test, in spite of being nonparametric and making no assumption on $P$ and $Q$ (the T-test is further advantaged by its excessive Type-I error). However, this is true only with optimization of the test locations. This is reflected in the test power of ME-grid in Fig. 3b which drops monotonically as dimension increases, highlighting the importance of test location optimization. The performance of MMD-lin degrades quickly with increasing dimension, as expected from Ramdas et al. (2015).

**4. Distinguishing articles from two categories**    We now turn to performance on real data. We first consider the problem of distinguishing two categories of publications at the conference on Neural Information Processing Systems (NIPS). Out of 5903 papers published in NIPS from 1988 to 2015, we manually select disjoint subsets related to Bayesian inference (Bayes), neuroscience (Neuro), deep learning (Deep), and statistical learning theory (Learn) (see Sec. B). Each paper is represented as a bag of words using TF-IDF (Manning et al., 2008) as features. We perform stemming, remove all stop words, and retain only nouns. A further filtering of document-frequency (DF) of words that satisfies $5 \leq \mathrm{DF} \leq 2000$ yields approximately 5000 words from which 2000 words (i.e., $d = 2000$ dimensions) are randomly selected. See Sec. B for more details on the preprocessing. For ME and SCF tests, we use only one test location i.e., set $J = 1$. We perform 1000 permutations to approximate the null distribution of MMD-quad in this and the following experiments.

Type-I errors and test powers are summarized in Table. 2. The first column indicates the categories of the papers in the two samples. In Bayes-Bayes problem, papers on Bayesian inference are randomly partitioned into two samples in each trial. This task represents a case in which $H_0$ holds. Among all the linear-time tests, we observe that ME-full has the highest test power in all the tasks, attaining a maximum test power of 1.0 in the Bayes-Neuro problem. This high performance assures that although different test locations $\mathcal{V}$ may be selected in different trials, these locations are each informative. It is interesting to observe that ME-full has performance close to or better than MMD-quad, which requires $O(n^2)$ runtime complexity. Besides clear advantages of interpretability and linear runtime of the proposed tests, these results suggest that evaluating the differences in expectations of analytic functions at particular locations can yield an equally powerful test at a much lower cost, as opposed to

Table 3: Type-I errors and powers in the problem of distinguishing positive (+) and negative (-) facial expressions. $\alpha = 0.01$. $J = 1$.

| Problem | $n^{te}$ | ME-full | ME-grid | SCF-full | SCF-grid | MMD-quad | MMD-lin |
|---|---|---|---|---|---|---|---|
| $\pm$ vs. $\pm$ | 201 | .010 | .012 | .014 | .002 | .018 | .008 |
| $+$ vs. $-$ | 201 | .998 | .656 | 1.00 | .750 | 1.00 | .578 |

computing the RKHS norm of the witness function as done in MMD. Unlike Blobs, however, Fourier features are less powerful in this setting.

We further investigate the interpretability of the ME test by the following procedure. For the learned test location $\mathbf{v}^t \in \mathbb{R}^d$ ($d = 2000$) in trial $t$, we construct $\tilde{\mathbf{v}}^t = (\tilde{v}_1^t, \ldots, \tilde{v}_d^t)$ such that $\tilde{v}_j^t = |v_j^t|$. Let $\eta_j^t \in \{0, 1\}$ be an indicator variable taking value 1 if $\tilde{v}_j^t$ is among the top five largest for all $j \in \{1, \ldots, d\}$, and 0 otherwise. Define $\eta_j := \sum_t \eta_j^t$ as a proxy indicating the significance of word $j$ i.e., $\eta_j$ is high if word $j$ is frequently among the top five largest as measured by $\tilde{v}_j^t$. The top seven words as sorted in descending order by $\eta_j$ in the Bayes-Neuro problem are *spike, markov, cortex, dropout, recurr, iii, gibb*, showing that the learned test locations are highly interpretable. Indeed, "markov" and "gibb" (i.e., stemmed from Gibbs) are discriminative terms in Bayesian inference category, and "spike" and "cortex" are key terms in neuroscience. We give full lists of discriminative terms learned in all the problems in Sec. B.1. To show that not all the randomly selected 2000 terms are informative, if the definition of $\eta_j^t$ is modified to consider the least important words (i.e., $\eta_j$ is high if word $j$ is frequently among the top five smallest as measured by $\tilde{v}_j^t$), we instead obtain *circumfer, bra, dominiqu, rhino, mitra, kid, impostor,* which are not discriminative.

**5. Distinguishing positive and negative emotions** In the final experiment, we study how well ME and SCF tests can distinguish two samples of photos of people showing positive and negative facial expressions. Our emphasis is on the discriminative features of the faces identified by ME test showing how the two groups differ. For this purpose, we use Karolinska Directed Emotional Faces (KDEF) dataset (Lundqvist et al., 1998) containing 5040 aligned face images of 70 amateur actors, 35 females and 35 males. We use only photos showing front views of the faces. In the dataset, each actor displays seven expressions: happy (HA), neutral (NE), surprised (SU), sad (SA), afraid (AF), angry (AN), and disgusted (DI). We assign HA, NE, and SU faces into the positive emotion group (i.e., samples from $P$), and AF, AN and DI faces into the negative emotion group (samples from $Q$). We denote this problem as "$+$ vs. $-$". Examples of six facial expressions from one actor are shown in Fig. 4. Photos of the SA group are unused to keep the sizes of the two samples

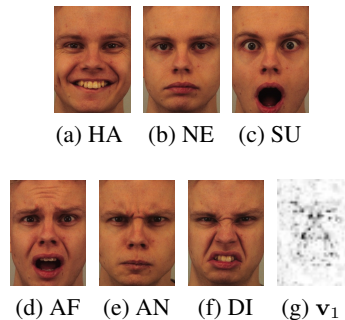

(a) HA    (b) NE    (c) SU

(d) AF    (e) AN    (f) DI    (g) $\mathbf{v}_1$

Figure 4: (a)-(f): Six facial expressions of actor AM05 in the KDEF data. (g): Average across trials of the learned test locations $\mathbf{v}_1$.

the same. Each image of size $562 \times 762$ pixels is cropped to exclude the background, resized to $48 \times 34 = 1632$ pixels ($d$), and converted to grayscale.

We run the tests 500 times with the same setting used previously i.e., Gaussian kernels, and $J = 1$. The type-I errors and test powers are shown in Table 3. In the table, "$\pm$ vs. $\pm$" is a problem in which all faces expressing the six emotions are randomly split into two samples of equal sizes i.e., $H_0$ is true. Both ME-full and SCF-full achieve high test powers while maintaining the correct type-I errors.

As a way to interpret how positive and negative emotions differ, we take an average across trials of the learned test locations of ME-full in the "$+$ vs. $-$" problem. This average is shown in Fig. 4g. We see that the test locations faithfully capture the difference of positive and negative emotions by giving more weights to the regions of nose, upper lip, and nasolabial folds (smile lines), confirming the interpretability of the test in a high-dimensional setting.

**Acknowledgement**

We thank the Gatsby Charitable Foundation for the financial support.

## Footnotes

[1]Chwialkowski et al. (2015) chooses the Gaussian width that minimizes the median of the p-values, a heuristic that does not directly address test power. Here, we perform a grid search to choose the best Gaussian width by maximizing $\hat{\lambda}_{n/2}^{tr}$ as done in ME-full and SCF-full.

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
