[Supplementary Material · fotest_nips2016_sup.pdf]

# Interpretable Distribution Features with Maximum Testing Power

## Supplementary Material

## A   Algorithm

The full algorithm for the proposed tests from parameter tuning to the actual two-sample testing is given in Algorithm 1.

---

**Algorithm 1** Optimizing parameters and testing

---

**Input:** Two samples $\mathsf{X}$, $\mathsf{Y}$, significance level $\alpha$, and number of test locations $J$

1: Split $\mathsf{D} := (\mathsf{X}, \mathsf{Y})$ into disjoint training and test sets, $\mathsf{D}^{tr}$ and $\mathsf{D}^{te}$, of the same size $n^{te}$.
2: Optimize parameters $\theta = \arg\max_\theta \hat{\lambda}^{tr}_{n/2}(\theta)$ where $\hat{\lambda}^{tr}_{n/2}(\theta)$ is computed with the training set $\mathsf{D}^{tr}$.
3: Set $T_\alpha$ to the $(1 - \alpha)$-quantile of $\chi^2(J')$.
4: Compute the test statistic $\hat{\lambda}^{te}_{n/2}(\theta)$ using $\mathsf{D}^{te}$.
5: Reject $H_0$ if $\hat{\lambda}^{te}_{n/2}(\theta) > T_\alpha$.

---

## B   Experiments on NIPS text collection

The full procedure for processing the NIPS text collection is summarized as following.

1. Download all 5903 papers from 1988 to 2015 from `https://papers.nips.cc/` as PDF files.

2. Convert each PDF file to text with `pdftotext`[2].

3. Remove all stop words. We use the list of stop words from `http://www.ranks.nl/stopwords`.

4. Keep only nouns. We use the list of nouns as available in WordNet-3.0[3].

5. Keep only words which contain only English alphabets i.e., does not contain punctuations or numbers. Also, word length must be between 3 and 20 characters (inclusive).

6. Keep only words which occur in at least 5 documents, and in no more than 2000 documents.

7. Convert all characters to small case. Stem all words with SnowballStemmer in NLTK (Bird et al., 2009). For example, "recognize" and "recognizer" become "recogn" after stemming.

8. Categorize papers into disjoint collections. A paper is treated as belonging to a group if its title has at least one word from the list of keywords for the category. Papers that match the criteria of both categories are not considered. The lists of keywords are as follows.

   (a) **Bayesian inference** (Bayes): graphical model, bayesian, inference, mcmc, monte carlo, posterior, prior, variational, markov, latent, probabilistic, exponential family.

   (b) **Deep learning** (Deep): deep, drop out, auto-encod, convolutional, neural net, belief net, boltzmann.

   (c) **Learning theory** (Learn): learning theory, consistency, theoretical guarantee, complexity, pac-bayes, pac-learning, generalization, uniform converg, bound, deviation, inequality, risk min, minimax, structural risk, VC, rademacher, asymptotic.

   (d) **Neuroscience** (Neuro): motor control, neural, neuron, spiking, spike, cortex, plasticity, neural decod, neural encod, brain imag, biolog, perception, cognitive, emotion, synap, neural population, cortical, firing rate, firing-rate, sensor.

9. Randomly select 2000 words from the remaining words.

10. Treat each paper as a bag of words and construct a feature vector with TF-IDF (Manning et al., 2008).

### B.1 Discriminative terms identified by ME test

In this section, we provide full lists of discriminative terms following the procedure described in Sec. 4. The top ten words in each problem are as follows.

- **Bayes-Bayes**: collabor, traffic, bay, permut, net, central, occlus, mask, draw, joint.
- **Bayes-Deep**: infer, bay, mont, adaptor, motif, haplotyp, ecg, covari, boltzmann, classifi.
- **Bayes-Learn**: infer, markov, graphic, segment, bandit, boundari, favor, carlo, prioriti, prop.
- **Bayes-Neuro**: spike, markov, cortex, dropout, recurr, iii, gibb, basin, circuit, subsystem.
- **Learn-Deep**: deep, forward, delay, subgroup, bandit, recept, invari, overlap, inequ, pia.
- **Learn-Neuro**: polici, interconnect, hardwar, decay, histolog, edg, period, basin, inject, human.

## C Runtimes

In this section, we provide runtimes of all the experiments. The runtimes of the "Test power vs. sample $n$" experiment are shown in Fig. 5. The runtimes of the "Test power vs. dimension $d$" experiment are shown in Fig. 6. Table 4, 5 give the runtimes of the two real-data experiments.

Figure 5: Plots of runtimes in the "Test power vs. sample $n$" experiment.

(a) SG. $d = 50$.  (b) GMD. $d = 100$.  (c) GVD. $d = 50$.  (d) Blobs.

(a) SG  (b) GMD  (c) GVD

Figure 6: Plots of runtimes in the "Test power vs. dimension $d$" experiment. The test sample size is $10000$.

Table 4: Runtimes (in seconds) in the problem of distinguishing NIPS papers from two categories.

| Problem | $n^{te}$ | ME-full | ME-grid | SCF-full | SCF-grid | MMD-quad | MMD-lin |
|---------|----------|---------|---------|----------|----------|----------|---------|
| Bayes-Bayes | 215 | 126.7 | 116 | 34.67 | 1.855 | 13.66 | .6112 |
| Bayes-Deep | 216 | 118.3 | 111.7 | 36.41 | 1.933 | 13.59 | .5105 |
| Bayes-Learn | 138 | 94.59 | 89.16 | 23.69 | 1.036 | 2.152 | .36 |
| Bayes-Neuro | 394 | 142.5 | 130.3 | 69.19 | 3.533 | 32.71 | .8643 |
| Learn-Deep | 149 | 105 | 99.59 | 24.99 | 1.253 | 2.417 | .4744 |
| Learn-Neuro | 146 | 101.2 | 93.53 | 25.29 | 1.178 | 2.351 | .3658 |

Table 5: Runtimes (in seconds) in the problem of distinguishing positive (+) and negative (-) facial expressions.

| Problem | $n^{te}$ | ME-full | ME-grid | SCF-full | SCF-grid | MMD-quad | MMD-lin |
|---------|----------|---------|---------|----------|----------|----------|---------|
| $\pm$ vs. $\pm$ | 201 | 87.7 | 83.4 | 10.5 | 1.45 | 9.93 | 0.464 |
| $+$ vs. $-$ | 201 | 85.0 | 80.6 | 11.7 | 1.42 | 10.4 | 0.482 |

In the cases where $n$ is large (Fig. 5), MMD-quad has the largest runtime due to its quadratic dependency on the sample size. In the extreme case where the test sample size is $10000$ (Fig. 6), it is computationally infeasible to run MMD-quad. We observe that the proposed ME-full and SCF-full have a slight overhead from the parameter optimization. However, since the optimization procedure is also linear in $n$, we are able to conduct an accurate test in less than 10 minutes even when the test sample size is $10000$ and $d = 1500$ (see Fig. 6a, 6b). We note that the actual tests (after optimization) for all ME and SCF variants take less than one second in all cases. In the ME-full, we initialize the test locations with realizations from two multivariate normal distributions fitted to samples from $P$ and $Q$. When $d$ is large, this heuristic can be expensive. An alternative initialization scheme for $\mathcal{V}$ is to randomly select $J$ points from the two samples.

# D   Proof of theorem 2

Recall Theorem 2:

**Theorem 2** (Consistency of $\hat{\lambda}_n$ in the ME test). *Let $\mathcal{X} \subseteq \mathbb{R}^d$ be a measurable set, and $\mathbb{V}$ be a collection in which each element is a set of $J$ test locations. All suprema over $\mathcal{V}$ and $k$ are to be understood as $\sup_{\mathcal{V} \in \mathbb{V}}$ and $\sup_{k \in \mathcal{K}}$ respectively. For a class of kernels $\mathcal{K}$ on $\mathcal{X} \subseteq \mathbb{R}^d$, define*

$$\mathcal{F}_1 := \{\mathbf{x} \mapsto k(\mathbf{x}, \mathbf{v}) \mid k \in \mathcal{K}, \mathbf{v} \in \mathcal{X}\}, \quad \mathcal{F}_2 := \{\mathbf{x} \mapsto k(\mathbf{x}, \mathbf{v})k(\mathbf{x}, \mathbf{v}') \mid k \in \mathcal{K}, \mathbf{v}, \mathbf{v}' \in \mathcal{X}\}, \quad (1)$$

$$\mathcal{F}_3 := \{(\mathbf{x}, \mathbf{y}) \mapsto k(\mathbf{x}, \mathbf{v})k(\mathbf{y}, \mathbf{v}') \mid k \in \mathcal{K}, \mathbf{v}, \mathbf{v}' \in \mathcal{X}\}. \quad (2)$$

*Assume that (1) $\mathcal{K}$ is a uniformly bounded (by $B$) family of $k : \mathcal{X} \times \mathcal{X} \to \mathbb{R}$ measurable kernels, (2) $\tilde{c} := \sup_{\mathcal{V},k} \|\mathbf{\Sigma}^{-1}\|_F < \infty$, and (3) $\mathcal{F}_i = \{f_{\theta_i} \mid \theta_i \in \Theta_i\}$ is VC-subgraph with VC-index $VC(\mathcal{F}_i)$, and $\theta \mapsto f_{\theta_i}(m)$ is continuous $(\forall m, i = 1, 2, 3)$. Let $\bar{c}_1 := 4B^2 J\sqrt{J}\tilde{c}, \bar{c}_2 := 4B\sqrt{J}\tilde{c}$, and $\bar{c}_3 := 4B^2 J\tilde{c}^2$. Let $C_i$-s $(i = 1, 2, 3)$ be the universal constants associated to $\mathcal{F}_i$-s according to Theorem 2.6.7 in van der Vaart and Wellner (2000). Then for any $\delta \in (0, 1)$ with probability at least $1 - \delta$,*

$$\sup_{\mathcal{V},k} \left| \bar{\mathbf{z}}_n^\top (\mathbf{S}_n + \gamma_n I)^{-1} \bar{\mathbf{z}}_n - \boldsymbol{\mu}^\top \mathbf{\Sigma}^{-1} \boldsymbol{\mu} \right|$$

$$\leq 2T_{\mathcal{F}_1} \left( \frac{2}{\gamma_n} \bar{c}_1 BJ \frac{2n - 1}{n - 1} + \bar{c}_2 \sqrt{J} \right) + \frac{2}{\gamma_n} \bar{c}_1 J(T_{\mathcal{F}_2} + T_{\mathcal{F}_3}) + \frac{8}{\gamma_n} \frac{\bar{c}_1 B^2 J}{n - 1} + \bar{c}_3 \gamma_n, \text{ where}$$

$$T_{\mathcal{F}_j} = \frac{16\sqrt{2}B^{\zeta_j}}{\sqrt{n}} \left( 2\sqrt{\log\left[C_j \times VC(\mathcal{F}_j)(16e)^{VC(\mathcal{F}_j)}\right]} + \frac{\sqrt{2\pi[VC(\mathcal{F}_j) - 1]}}{2} \right) + B^{\zeta_j}\sqrt{\frac{2\log(5/\delta)}{n}},$$

*for $j = 1, 2, 3$ and $\zeta_1 = 1, \zeta_2 = \zeta_3 = 2$.*

A proof is given as follows.

## D.1   Notations

Let $\langle \mathbf{A}, \mathbf{B} \rangle_F := \text{tr}\left(\mathbf{A}^\top \mathbf{B}\right)$ be the Frobenius inner product, and $\|\mathbf{A}\|_F := \sqrt{\langle \mathbf{A}, \mathbf{A} \rangle_F}$. $\mathbf{A} \succeq \mathbf{0}$ means that $\mathbf{A} \in \mathbb{R}^{d \times d}$ is symmetric, positive semidefinite. For $\mathbf{a} \in \mathbb{R}^d$, $\|\mathbf{a}\|_2 = \langle \mathbf{a}, \mathbf{a} \rangle_2 = \mathbf{a}^\top \mathbf{a}$. $[\mathbf{a}_1; \ldots; \mathbf{a}_N] \in \mathbb{R}^{d_1 + \ldots + d_N}$ is the concatenation of the $\mathbf{a}_n \in \mathbb{R}^{d_n}$ vectors. $\mathbb{R}^+$ is the set of positive reals. $f \circ g$ is the composition of function $f$ and $g$. Let $\mathcal{M}$ denote a general metric space below. In measurability requirements metric spaces are meant to be endowed with their Borel $\sigma$-algebras.

Let $\mathcal{C}$ be a collection of subsets of $\mathcal{M}$ $(C \subseteq 2^{\mathcal{M}})$. $\mathcal{C}$ is said to shatter an $\{p_1, p_2, \ldots, p_i\} \subseteq \mathcal{M}$ set, if for any $S \subseteq \{p_1, p_2, \ldots, p_i\}$ there exist $C \in \mathcal{C}$ such that $S = C \cap \{p_1, p_2, \ldots, p_i\}$; in other words, arbitrary subset of $\{p_1, p_2, \ldots, p_i\}$ can be cut out by an element of $\mathcal{C}$. The VC index of $\mathcal{C}$ is the smallest $i$ for which no set of size i is shattered:

$$VC(\mathcal{C}) = \inf\left\{i : \max_{p_1, \ldots, p_i} |\{C \cap \{p_1, \ldots, p_i\} : C \in \mathcal{C}\}| < 2^i\right\}.$$

A collection $\mathcal{C}$ of measurable sets is called VC-class if its index $VC(\mathcal{C})$ is finite. The subgraph of a real-valued function $f : \mathcal{M} \to \mathbb{R}$ is $sub(f) = \{(m, u) : u < f(m)\} \subseteq \mathcal{M} \times \mathbb{R}$. A collection of $\mathcal{F}$ measurable functions is called VC-subgraph class, or shortly VC if the collection of all subgraphs of $\mathcal{F}$, $\{sub(f)\}_{f \in \mathcal{F}}$ is a VC-class of sets; its index is defined as $VC(\mathcal{F}) := VC\left(\{sub(f)\}_{f \in \mathcal{F}}\right)$.

Let $L^0(\mathcal{M})$ be the set of $\mathcal{M} \to \mathbb{R}$ measurable functions. Given an i.i.d. (independent identically distributed) sample from $\mathbb{P}$ ($w_i \overset{i.i.d.}{\sim} \mathbb{P}$), let $w_{1:n} = (w_1, \ldots, w_n)$ and let $\mathbb{P}_n = \frac{1}{n}\sum_{i=1}^n \delta_{w_i}$ denote the empirical measure. $L^q(\mathcal{M}, \mathbb{P}_n) = \left\{f \in L^0(\mathcal{M}) : \|f\|_{L^q(\mathcal{M}, \mathbb{P}_n)} := \left[\int_{\mathcal{M}} |f(w)|^q d\mathbb{P}_n(w)\right]^{\frac{1}{q}} = \left[\frac{1}{n}\sum_{i=1}^n |f(w_i)|^q\right]^{\frac{1}{q}} < \infty\right\}$ ($1 \leq q < \infty$), $\|f\|_{L^\infty(\mathcal{M})} := \sup_{m \in \mathcal{M}} |f(m)|$. Define $\mathbb{P}f := \int_{\mathcal{M}} f(w) d\mathbb{P}(w)$, where $\mathbb{P}$ is a probability distribution on $\mathcal{M}$. Let $\|\mathbb{P}_n - \mathbb{P}\|_{\mathcal{F}} := \sup_{f \in \mathcal{F}} |\mathbb{P}_n f - \mathbb{P}f|$.

The diameter of a class $\mathcal{F} \subseteq L^2(\mathcal{M}, \mathbb{P}_n)$ is $\text{diam}(\mathcal{F}, L^2(\mathcal{M}, \mathbb{P}_n)) := \sup_{f, f' \in \mathcal{F}} \|f - f'\|_{L^2(\mathcal{M}, \mathbb{P}_n)}$, its $r$-covering number ($r > 0$) is the size of the smallest $r$-net

$$N\left(r, \mathcal{F}, L^2(\mathcal{M}, \mathbb{P}_n)\right) = \inf\left\{t \geq 1 : \exists f_1, \ldots, f_t \in \mathcal{F} \text{ such that } \mathcal{F} \subseteq \cup_{i=1}^t B(r, f_i)\right\},$$

where $B(r, f) = \{g \in L^2(\mathcal{M}, \mathbb{P}_n) \mid \|f - g\|_{L^2(\mathcal{M}, \mathbb{P}_n)} \le r\}$ is the ball with center $f$ and radius $r$. $\times_{i=1}^N \mathbb{Q}_i$ is the $N$-fold product measure. For sets $Q_i$, $\times_{i=1}^n Q_i$ is their Cartesian product. For a function class $\mathcal{F} \subseteq L^0(\mathcal{M})$ and $w_{1:n} \in \mathcal{M}^n$, $R(\mathcal{F}, w_{1:n}) := \mathbb{E}_{\mathbf{r}}\left[\sup_{f \in \mathcal{F}} \left|\frac{1}{n}\sum_{i=1}^n r_i f(w_i)\right|\right]$ is the empirical Rademacher average, where $\mathbf{r} := r_{1:n}$ and $r_i$-s are i.i.d. samples from a Rademacher random variable [$\mathbb{P}(r_i = 1) = \mathbb{P}(r_i = -1) = \frac{1}{2}$]. Let $(\Theta, \rho)$ be a metric space; a collection of $\mathcal{F} = \{f_\theta \mid \theta \in \Theta\} \subseteq L^0(\mathcal{M})$ functions is called a separable Carathéodory family if $\Theta$ is separable and $\theta \mapsto f_\theta(m)$ is continuous for all $m \in \mathcal{M}$. $span(\cdot)$ denotes the linear hull of its arguments. $\Gamma(t) = \int_0^\infty u^{t-1} e^{-u} \mathrm{d}u$ denotes the Gamma function.

## D.2 Bound in terms of $\mathbf{S}_n$ and $\bar{\mathbf{z}}_n$

For brevity, we will interchangeably use $\mathbf{S}_n$ for $\mathbf{S}_n(\mathcal{V})$ and $\bar{\mathbf{z}}_n$ for $\bar{\mathbf{z}}_n(\mathcal{V})$. $\mathbf{S}_n(\mathcal{V})$ and $\bar{\mathbf{z}}_n(\mathcal{V})$ will be used mainly when the dependency of $\mathcal{V}$ needs to be emphasized. We start with $\sup_{\mathcal{V},k} \left|\bar{\mathbf{z}}_n^\top (\mathbf{S}_n + \gamma_n I)^{-1} \bar{\mathbf{z}}_n - \boldsymbol{\mu}^\top \boldsymbol{\Sigma}^{-1} \boldsymbol{\mu}\right|$ and upper bound the argument of $\sup_{\mathcal{V},k}$ as

$$\left|\bar{\mathbf{z}}_n^\top (\mathbf{S}_n + \gamma_n I)^{-1} \bar{\mathbf{z}}_n - \boldsymbol{\mu}^\top \boldsymbol{\Sigma}^{-1} \boldsymbol{\mu}\right|$$
$$= \left|\bar{\mathbf{z}}_n^\top (\mathbf{S}_n + \gamma_n I)^{-1} \bar{\mathbf{z}}_n - \boldsymbol{\mu}^\top (\boldsymbol{\Sigma} + \gamma_n I)^{-1} \boldsymbol{\mu} + \boldsymbol{\mu}^\top (\boldsymbol{\Sigma} + \gamma_n I)^{-1} \boldsymbol{\mu} - \boldsymbol{\mu}^\top \boldsymbol{\Sigma}^{-1} \boldsymbol{\mu}\right|$$
$$\le \left|\bar{\mathbf{z}}_n^\top (\mathbf{S}_n + \gamma_n I)^{-1} \bar{\mathbf{z}}_n - \boldsymbol{\mu}^\top (\boldsymbol{\Sigma} + \gamma_n I)^{-1} \boldsymbol{\mu}\right| + \left|\boldsymbol{\mu}^\top (\boldsymbol{\Sigma} + \gamma_n I)^{-1} \boldsymbol{\mu} - \boldsymbol{\mu}^\top \boldsymbol{\Sigma}^{-1} \boldsymbol{\mu}\right|$$
$$:= (\square_1) + (\square_2).$$

For $(\square_1)$, we have

$$\left|\bar{\mathbf{z}}_n^\top (\mathbf{S}_n + \gamma_n I)^{-1} \bar{\mathbf{z}}_n - \boldsymbol{\mu}^\top (\boldsymbol{\Sigma} + \gamma_n I)^{-1} \boldsymbol{\mu}\right|$$
$$= \left|\left\langle \bar{\mathbf{z}}_n \bar{\mathbf{z}}_n^\top, (\mathbf{S}_n + \gamma_n I)^{-1}\right\rangle_F - \left\langle \boldsymbol{\mu}\boldsymbol{\mu}^\top, (\boldsymbol{\Sigma} + \gamma_n I)^{-1}\right\rangle_F\right|$$
$$= \left|\left\langle \bar{\mathbf{z}}_n \bar{\mathbf{z}}_n^\top, (\mathbf{S}_n + \gamma_n I)^{-1}\right\rangle_F - \left\langle \bar{\mathbf{z}}_n \bar{\mathbf{z}}_n^\top, (\boldsymbol{\Sigma} + \gamma_n I)^{-1}\right\rangle_F + \left\langle \bar{\mathbf{z}}_n \bar{\mathbf{z}}_n^\top, (\boldsymbol{\Sigma} + \gamma_n I)^{-1}\right\rangle_F - \left\langle \boldsymbol{\mu}\boldsymbol{\mu}^\top, (\boldsymbol{\Sigma} + \gamma_n I)^{-1}\right\rangle_F\right|$$
$$\le \left|\left\langle \bar{\mathbf{z}}_n \bar{\mathbf{z}}_n^\top, (\mathbf{S}_n + \gamma_n I)^{-1} - (\boldsymbol{\Sigma} + \gamma_n I)^{-1}\right\rangle_F\right| + \left|\left\langle \bar{\mathbf{z}}_n \bar{\mathbf{z}}_n^\top - \boldsymbol{\mu}\boldsymbol{\mu}^\top, (\boldsymbol{\Sigma} + \gamma_n I)^{-1}\right\rangle_F\right|$$
$$= \|\bar{\mathbf{z}}_n \bar{\mathbf{z}}_n^\top\|_F \|(\mathbf{S}_n + \gamma_n I)^{-1} - (\boldsymbol{\Sigma} + \gamma_n I)^{-1}\|_F + \|\bar{\mathbf{z}}_n \bar{\mathbf{z}}_n^\top - \boldsymbol{\mu}\boldsymbol{\mu}^\top\|_F \|(\boldsymbol{\Sigma} + \gamma_n I)^{-1}\|_F$$
$$\overset{(a)}{\le} \|\bar{\mathbf{z}}_n \bar{\mathbf{z}}_n^\top\|_F \|(\mathbf{S}_n + \gamma_n I)^{-1}[(\boldsymbol{\Sigma} + \gamma_n I) - (\mathbf{S}_n + \gamma_n I)](\boldsymbol{\Sigma} + \gamma_n I)^{-1}\|_F + \|\bar{\mathbf{z}}_n \bar{\mathbf{z}}_n^\top - \bar{\mathbf{z}}_n \boldsymbol{\mu}^\top + \bar{\mathbf{z}}_n \boldsymbol{\mu}^\top - \boldsymbol{\mu}\boldsymbol{\mu}^\top\|_F \|\boldsymbol{\Sigma}^{-1}\|_F$$
$$\overset{(a)}{\le} \|\bar{\mathbf{z}}_n \bar{\mathbf{z}}_n^\top\|_F \|(\mathbf{S}_n + \gamma_n I)^{-1}\|_F \|\boldsymbol{\Sigma} - \mathbf{S}_n\|_F \|\boldsymbol{\Sigma}^{-1}\|_F + \|\bar{\mathbf{z}}_n(\bar{\mathbf{z}}_n - \boldsymbol{\mu})^\top\|_F \|\boldsymbol{\Sigma}^{-1}\|_F + \|(\bar{\mathbf{z}}_n - \boldsymbol{\mu})\boldsymbol{\mu}^\top\|_F \|\boldsymbol{\Sigma}^{-1}\|_F$$
$$\overset{(b)}{\le} \frac{\sqrt{J}}{\gamma_n} \|\bar{\mathbf{z}}_n\|_2^2 \|\boldsymbol{\Sigma} - \mathbf{S}_n\|_F \|\boldsymbol{\Sigma}^{-1}\|_F + \|\bar{\mathbf{z}}_n\|_2 \|\bar{\mathbf{z}}_n - \boldsymbol{\mu}\|_2 \|\boldsymbol{\Sigma}^{-1}\|_F + \|\boldsymbol{\mu}\|_2 \|\bar{\mathbf{z}}_n - \boldsymbol{\mu}\|_2 \|\boldsymbol{\Sigma}^{-1}\|_F,$$

where at (a) we use $\|(\boldsymbol{\Sigma} + \gamma_n I)^{-1}\|_F \le \|\boldsymbol{\Sigma}^{-1}\|_F$ and at (b) we use $\|(\mathbf{S}_n + \gamma_n I)^{-1}\|_F \le \sqrt{J}\|(\mathbf{S}_n + \gamma_n I)^{-1}\|_2 \le \sqrt{J}/\gamma_n$.

For $(\square_2)$, we have

$$\left|\boldsymbol{\mu}^\top (\boldsymbol{\Sigma} + \gamma_n I)^{-1} \boldsymbol{\mu} - \boldsymbol{\mu}^\top \boldsymbol{\Sigma}^{-1} \boldsymbol{\mu}\right| = \left|\left\langle \boldsymbol{\mu}\boldsymbol{\mu}^\top, (\boldsymbol{\Sigma} + \gamma_n I)^{-1} - \boldsymbol{\Sigma}^{-1}\right\rangle_F\right|$$
$$\le \|\boldsymbol{\mu}\boldsymbol{\mu}^\top\|_F \|(\boldsymbol{\Sigma} + \gamma_n I)^{-1} - \boldsymbol{\Sigma}^{-1}\|_F$$
$$= \|\boldsymbol{\mu}\|_2^2 \|(\boldsymbol{\Sigma} + \gamma_n I)^{-1}[\boldsymbol{\Sigma} - (\boldsymbol{\Sigma} + \gamma_n I)]\boldsymbol{\Sigma}^{-1}\|_F$$
$$= \gamma_n \|\boldsymbol{\mu}\|_2^2 \|(\boldsymbol{\Sigma} + \gamma_n I)^{-1}\boldsymbol{\Sigma}^{-1}\|_F$$
$$\le \gamma_n \|\boldsymbol{\mu}\|_2^2 \|(\boldsymbol{\Sigma} + \gamma_n I)^{-1}\|_F \|\boldsymbol{\Sigma}^{-1}\|_F$$
$$\overset{(a)}{\le} \gamma_n \|\boldsymbol{\mu}\|_2^2 \|\boldsymbol{\Sigma}^{-1}\|_F^2.$$

Combining the upper bounds for $(\square_1)$ and $(\square_2)$, we arrive at

$$\left|\bar{\mathbf{z}}_n^\top (\mathbf{S}_n + \gamma_n I)^{-1} \bar{\mathbf{z}}_n - \boldsymbol{\mu}^\top \boldsymbol{\Sigma}^{-1} \boldsymbol{\mu}\right|$$
$$\le \frac{\sqrt{J}}{\gamma_n} \|\bar{\mathbf{z}}_n\|_2^2 \|\boldsymbol{\Sigma} - \mathbf{S}_n\|_F \|\boldsymbol{\Sigma}^{-1}\|_F + (\|\bar{\mathbf{z}}_n\|_2 + \|\boldsymbol{\mu}\|_2)\|\bar{\mathbf{z}}_n - \boldsymbol{\mu}\|_2 \|\boldsymbol{\Sigma}^{-1}\|_F + \gamma_n \|\boldsymbol{\mu}\|_2^2 \|\boldsymbol{\Sigma}^{-1}\|_F^2$$

$$\leq 4B^2 J\tilde{c}\frac{\sqrt{J}}{\gamma_n}\|\boldsymbol{\Sigma} - \mathbf{S}_n\|_F + 4B\sqrt{J}\tilde{c}\|\bar{\mathbf{z}}_n - \boldsymbol{\mu}\|_2 + 4B^2 J\tilde{c}^2\gamma_n$$

$$= \frac{\bar{c}_1}{\gamma_n}\|\boldsymbol{\Sigma} - \mathbf{S}_n\|_F + \bar{c}_2\|\bar{\mathbf{z}}_n - \boldsymbol{\mu}\|_2 + \bar{c}_3\gamma_n \tag{3}$$

with $\bar{c}_1 := 4B^2 J\sqrt{J}\tilde{c}, \bar{c}_2 := 4B\sqrt{J}\tilde{c}, \bar{c}_3 := 4B^2 J\tilde{c}^2$, and $\tilde{c} := \sup_{\mathcal{V},k}\|\boldsymbol{\Sigma}^{-1}\|_F < \infty$, where we applied the triangle inequality, the CBS (Cauchy-Bunyakovskii-Schwarz) inequality, and $\|\mathbf{ab}^\top\|_F = \|\mathbf{a}\|_2\|\mathbf{b}\|_2$. The boundedness of kernel $k$ with the Jensen inequality implies that

$$\|\bar{\mathbf{z}}_n\|_2^2 = \|\frac{1}{n}\sum_{i=1}^n\mathbf{z}_i\|_2^2 \leq \frac{1}{n}\sum_{i=1}^n\|\mathbf{z}_i\|_2^2 = \frac{1}{n}\sum_{i=1}^n\|(k(\mathbf{x}_i,\mathbf{v}_j) - k(\mathbf{y}_i,\mathbf{v}_j))_{j=1}^J\|_2^2 \tag{4}$$

$$= \frac{1}{n}\sum_{i=1}^n\sum_{j=1}^J [k(\mathbf{x}_i,\mathbf{v}_j) - k(\mathbf{y}_i,\mathbf{v}_j)]^2$$

$$\leq \frac{2}{n}\sum_{i=1}^n\sum_{j=1}^J k^2(\mathbf{x}_i,\mathbf{v}_j) + k^2(\mathbf{y}_i,\mathbf{v}_j) \leq 4B^2 J, \tag{5}$$

$$\|\boldsymbol{\mu}(\mathcal{V})\|_2^2 = \sum_{j=1}^J \left(\mathbb{E}_{\mathbf{xy}}\left[k(\mathbf{x},\mathbf{v}_j) - k(\mathbf{y},\mathbf{v}_j)\right]\right)^2 \leq \sum_{j=1}^J \mathbb{E}_{\mathbf{xy}}\left[k(\mathbf{x},\mathbf{v}_j) - k(\mathbf{y},\mathbf{v}_j)\right]^2 \leq 4B^2 J. \tag{6}$$

Taking sup in (3), we get

$$\sup_{\mathcal{V},k}\left|\bar{\mathbf{z}}_n^\top(\mathbf{S}_n + \gamma_n I)^{-1}\bar{\mathbf{z}}_n - \boldsymbol{\mu}^\top\boldsymbol{\Sigma}^{-1}\boldsymbol{\mu}\right| \leq \frac{\bar{c}_1}{\gamma_n}\sup_{\mathcal{V},k}\|\boldsymbol{\Sigma} - \mathbf{S}_n\|_F + \bar{c}_2\sup_{\mathcal{V},k}\|\bar{\mathbf{z}}_n - \boldsymbol{\mu}\|_2 + \bar{c}_3\gamma_n.$$

### D.3 Empirical process bound on $\bar{\mathbf{z}}_n$

Recall that $\bar{\mathbf{z}}_n(\mathcal{V}) = \frac{1}{n}\sum_{i=1}^n\mathbf{z}_i(\mathcal{V}) \in \mathbb{R}^J$, $\mathbf{z}_i(\mathcal{V}) = (k(\mathbf{x}_i,\mathbf{v}_j) - k(\mathbf{y}_i,\mathbf{v}_j))_{j=1}^J \in \mathbb{R}^J$, $\boldsymbol{\mu}(\mathcal{V}) = (\mathbb{E}_{\mathbf{xy}}[k(\mathbf{x},\mathbf{v}_j) - k(\mathbf{y},\mathbf{v}_j)])_{j=1}^J$; thus

$$\sup_{\mathcal{V}}\sup_{k\in\mathcal{K}}\|\bar{\mathbf{z}}_n(\mathcal{V}) - \boldsymbol{\mu}(\mathcal{V})\|_2 = \sup_{\mathcal{V}}\sup_{k\in\mathcal{K}}\sup_{\mathbf{b}\in B(1,\mathbf{0})}\langle\mathbf{b},\bar{\mathbf{z}}_n(\mathcal{V}) - \boldsymbol{\mu}(\mathcal{V})\rangle_2$$

using that $\|\mathbf{a}\|_2 = \sup_{\mathbf{b}\in B(1,\mathbf{0})}\langle\mathbf{a},\mathbf{b}\rangle_2$. Let us bound the argument of the supremum:

$$\langle\mathbf{b},\bar{\mathbf{z}}_n(\mathcal{V}) - \boldsymbol{\mu}(\mathcal{V})\rangle_2 \leq \sum_{j=1}^J |b_j|\left|\frac{1}{n}\sum_{i=1}^n[k(\mathbf{x}_i,\mathbf{v}_j) - k(\mathbf{y}_i,\mathbf{v}_j)] - \mathbb{E}_{\mathbf{xy}}\left[k(\mathbf{x},\mathbf{v}_j) - k(\mathbf{y},\mathbf{v}_j)\right]\right|$$

$$\leq \sum_{j=1}^J |b_j|\left(\left|\frac{1}{n}\sum_{i=1}^n k(\mathbf{x}_i,\mathbf{v}_j) - \mathbb{E}_{\mathbf{x}}k(\mathbf{x},\mathbf{v}_j)\right| + \left|\frac{1}{n}\sum_{i=1}^n k(\mathbf{y}_i,\mathbf{v}_j) - \mathbb{E}_{\mathbf{y}}k(\mathbf{y},\mathbf{v}_j)\right|\right)$$

$$\leq \sqrt{J}\sup_{\mathbf{v}\in\mathcal{X}}\sup_{k\in\mathcal{K}}\left|\frac{1}{n}\sum_{i=1}^n k(\mathbf{x}_i,\mathbf{v}) - \mathbb{E}_{\mathbf{x}}k(\mathbf{x},\mathbf{v})\right| + \sqrt{J}\sup_{\mathbf{v}\in\mathcal{X}}\sup_{k\in\mathcal{K}}\left|\frac{1}{n}\sum_{i=1}^n k(\mathbf{y}_i,\mathbf{v}) - \mathbb{E}_{\mathbf{y}}k(\mathbf{y},\mathbf{v})\right|$$

$$= \sqrt{J}\|P_n - P\|_{\mathcal{F}_1} + \sqrt{J}\|Q_n - Q\|_{\mathcal{F}_1} \tag{7}$$

by the triangle inequality and exploiting that $\|\mathbf{b}\|_1 \leq \sqrt{J}\|\mathbf{b}\|_2 \leq \sqrt{J}$ with $\mathbf{b} \in B(1,\mathbf{0})$. Thus, we have

$$\sup_{\mathcal{V}}\sup_{k\in\mathcal{K}}\|\bar{\mathbf{z}}_n(\mathcal{V}) - \boldsymbol{\mu}(\mathcal{V})\|_2 \leq \sqrt{J}\|P_n - P\|_{\mathcal{F}_1} + \sqrt{J}\|Q_n - Q\|_{\mathcal{F}_1}.$$

### D.4 Empirical process bound on $\mathbf{S}_n$

Noting that

$$\boldsymbol{\Sigma}(\mathcal{V}) = \mathbb{E}_{\mathbf{xy}}\left[\mathbf{z}(\mathcal{V})\mathbf{z}^\top(\mathcal{V})\right] - \boldsymbol{\mu}(\mathcal{V})\boldsymbol{\mu}^\top(\mathcal{V}), \qquad \mathbf{S}_n(\mathcal{V}) = \frac{1}{n}\sum_{a=1}^n\mathbf{z}_a(\mathcal{V})\mathbf{z}_a^\top(\mathcal{V}) - \frac{1}{n(n-1)}\sum_{a=1}^n\sum_{b\neq a}\mathbf{z}_a\mathbf{z}_b^T,$$

$$\mathbb{E}_{\mathbf{xy}}\left[\mathbf{z}(\mathcal{V})\mathbf{z}^\top(\mathcal{V})\right] = \mathbb{E}_{\mathbf{xy}}\left[\frac{1}{n}\sum_{a=1}^{n}\mathbf{z}_a(\mathcal{V})\mathbf{z}_a^\top(\mathcal{V})\right], \qquad \boldsymbol{\mu}(\mathcal{V})\boldsymbol{\mu}^\top(\mathcal{V}) = \mathbb{E}_{\mathbf{xy}}\left[\frac{1}{n(n-1)}\sum_{a=1}^{n}\sum_{b\neq a}\mathbf{z}_a(\mathcal{V})\mathbf{z}_b^T(\mathcal{V})\right],$$

we bound our target quantity as

$$\|\mathbf{S}_n(\mathcal{V}) - \boldsymbol{\Sigma}(\mathcal{V})\|_F \leq \left\|\frac{1}{n}\sum_{a=1}^{n}\mathbf{z}_a(\mathcal{V})\mathbf{z}_a^\top(\mathcal{V}) - \mathbb{E}_{\mathbf{xy}}\left[\mathbf{z}(\mathcal{V})\mathbf{z}^\top(\mathcal{V})\right]\right\|_F + \left\|\frac{1}{n(n-1)}\sum_{a=1}^{n}\sum_{b\neq a}\mathbf{z}_a(\mathcal{V})\mathbf{z}_b^T(\mathcal{V}) - \boldsymbol{\mu}(\mathcal{V})\boldsymbol{\mu}^\top(\mathcal{V})\right\|_F$$

$$=: (*_1) + (*_2). \tag{8}$$

$$(*_2) = \left\|\frac{1}{n}\sum_{a=1}^{n}\mathbf{z}_a(\mathcal{V})\left[\frac{1}{n-1}\sum_{b\neq a}\mathbf{z}_b^\top(\mathcal{V})\right] - \boldsymbol{\mu}(\mathcal{V})\boldsymbol{\mu}^\top(\mathcal{V})\right\|_F$$

$$\leq \left\|\frac{1}{n}\sum_{a=1}^{n}\mathbf{z}_a(\mathcal{V})\left(\frac{1}{n-1}\sum_{b\neq a}\mathbf{z}_b^\top(\mathcal{V}) - \boldsymbol{\mu}^\top(\mathcal{V})\right)\right\|_F + \left\|\left(\frac{1}{n}\sum_{a=1}^{n}\mathbf{z}_a(\mathcal{V}) - \boldsymbol{\mu}(\mathcal{V})\right)\boldsymbol{\mu}^\top(\mathcal{V})\right\|_F$$

$$\leq \left\|\left(\frac{1}{n}\sum_{a=1}^{n}\mathbf{z}_a(\mathcal{V})\right)\left(\frac{1}{n-1}\sum_{b=1}^{n}\mathbf{z}_b(\mathcal{V}) - \boldsymbol{\mu}(\mathcal{V})\right)^\top\right\|_F + \left\|\left(\frac{1}{n}\sum_{a=1}^{n}\mathbf{z}_a(\mathcal{V})\right)\frac{\mathbf{z}_a^\top(\mathcal{V})}{n-1}\right\|_F$$

$$+ \left\|\left(\frac{1}{n}\sum_{a=1}^{n}\mathbf{z}_a(\mathcal{V}) - \boldsymbol{\mu}(\mathcal{V})\right)\boldsymbol{\mu}^\top(\mathcal{V})\right\|_F$$

$$= \|\bar{\mathbf{z}}_n(\mathcal{V})\|_2\left\|\frac{1}{n-1}\sum_{b=1}^{n}\mathbf{z}_b(\mathcal{V}) - \boldsymbol{\mu}(\mathcal{V})\right\|_2 + \frac{1}{n-1}\|\bar{\mathbf{z}}_n(\mathcal{V})\|_2\|\mathbf{z}_a(\mathcal{V})\|_2 + \|\bar{\mathbf{z}}_n(\mathcal{V}) - \boldsymbol{\mu}(\mathcal{V})\|_2\|\boldsymbol{\mu}(\mathcal{V})\|_2$$

$$\leq 2B\sqrt{J}\left(\frac{n}{n-1}\|\bar{\mathbf{z}}_n - \boldsymbol{\mu}(\mathcal{V})\|_2 + \frac{2B\sqrt{J}}{n-1}\right) + \frac{1}{n-1}4B^2J + 2B\sqrt{J}\|\bar{\mathbf{z}}_n(\mathcal{V}) - \boldsymbol{\mu}(\mathcal{V})\|_2$$

$$= \frac{8B^2J}{n-1} + 2B\sqrt{J}\frac{2n-1}{n-1}\|\bar{\mathbf{z}}_n - \boldsymbol{\mu}(\mathcal{V})\|_2 \tag{9}$$

using the triangle inequality, the sub-additivity of sup, $\left\|\mathbf{a}\mathbf{b}^T\right\|_F = \|\mathbf{a}\|_2\|\mathbf{b}\|_2$, $\|\bar{\mathbf{z}}_n(\mathcal{V})\|_2 \leq 2B\sqrt{J}$, $\|\mathbf{z}_a(\mathcal{V})\|_2 \leq 2B\sqrt{J}$ [see Eq. (5)] and

$$\left\|\frac{1}{n-1}\sum_{b=1}^{n}\mathbf{z}_b(\mathcal{V}) - \boldsymbol{\mu}(\mathcal{V})\right\|_2 = \left\|\frac{n}{n-1}\bar{\mathbf{z}}_n - \frac{n}{n-1}\boldsymbol{\mu}(\mathcal{V}) + \frac{1}{n-1}\boldsymbol{\mu}(\mathcal{V})\right\|_2 \leq \frac{n}{n-1}\|\bar{\mathbf{z}}_n - \boldsymbol{\mu}(\mathcal{V})\|_2 + \frac{1}{n-1}\|\boldsymbol{\mu}(\mathcal{V})\|_2$$

with Eq. (6). Considering the first term in Eq. (8)

$$\left\|\frac{1}{n}\sum_{a=1}^{n}\mathbf{z}_a(\mathcal{V})\mathbf{z}_a^\top(\mathcal{V}) - \mathbb{E}_{\mathbf{xy}}\left[\mathbf{z}(\mathcal{V})\mathbf{z}^\top(\mathcal{V})\right]\right\|_F = \sup_{\mathbf{B}\in B(1,\mathbf{0})}\left\langle\mathbf{B}, \frac{1}{n}\sum_{a=1}^{n}\mathbf{z}_a(\mathcal{V})\mathbf{z}_a^\top(\mathcal{V}) - \mathbb{E}_{\mathbf{xy}}\left[\mathbf{z}(\mathcal{V})\mathbf{z}^\top(\mathcal{V})\right]\right\rangle_F$$

$$\leq \sup_{\mathbf{B}\in B(1,\mathbf{0})}\sum_{i,j=1}^{J}|B_{ij}|\left|\frac{1}{n}\sum_{a=1}^{n}[k(\mathbf{x}_a,\mathbf{v}_i) - k(\mathbf{y}_a,\mathbf{v}_i)][k(\mathbf{x}_a,\mathbf{v}_j) - k(\mathbf{y}_a,\mathbf{v}_j)] - \mathbb{E}_{\mathbf{xy}}\left([k(\mathbf{x},\mathbf{v}_i) - k(\mathbf{y},\mathbf{v}_i)][k(\mathbf{x},\mathbf{v}_j) - k(\mathbf{y},\mathbf{v}_j)]\right)\right|$$

$$\leq \sup_{\mathbf{B}\in B(1,\mathbf{0})}\sum_{i,j=1}^{J}|B_{ij}|\left(\left|\frac{1}{n}\sum_{a=1}^{n}k(\mathbf{x}_a,\mathbf{v}_i)k(\mathbf{x}_a,\mathbf{v}_j) - \mathbb{E}_{\mathbf{x}}\left[k(\mathbf{x},\mathbf{v}_i)k(\mathbf{x},\mathbf{v}_j)\right]\right|\right.$$

$$+ \left|\frac{1}{n}\sum_{a=1}^{n}k(\mathbf{x}_a,\mathbf{v}_i)k(\mathbf{y}_a,\mathbf{v}_j) - \mathbb{E}_{\mathbf{xy}}\left[k(\mathbf{x},\mathbf{v}_i)k(\mathbf{y},\mathbf{v}_j)\right]\right|$$

$$+ \left|\frac{1}{n}\sum_{a=1}^{n}k(\mathbf{y}_a,\mathbf{v}_i)k(\mathbf{x}_a,\mathbf{v}_j) - \mathbb{E}_{\mathbf{xy}}\left[k(\mathbf{y},\mathbf{v}_i)k(\mathbf{x},\mathbf{v}_j)\right]\right| + \left|\frac{1}{n}\sum_{a=1}^{n}k(\mathbf{y}_a,\mathbf{v}_i)k(\mathbf{y}_a,\mathbf{v}_j) - \mathbb{E}_{\mathbf{y}}\left[k(\mathbf{y},\mathbf{v}_i)k(\mathbf{y},\mathbf{v}_j)\right]\right|\right)$$

$$\leq J\sup_{\mathbf{v},\mathbf{v}'\in\mathcal{X}}\sup_{k\in\mathcal{K}}\left|\frac{1}{n}\sum_{a=1}^{n}k(\mathbf{x}_a,\mathbf{v})k(\mathbf{x}_a,\mathbf{v}') - \mathbb{E}_{\mathbf{x}}\left[k(\mathbf{x},\mathbf{v})k(\mathbf{x},\mathbf{v}')\right]\right|$$

$$+ 2J \sup_{\mathbf{v},\mathbf{v}'\in\mathcal{X}} \sup_{k\in\mathcal{K}} \left| \frac{1}{n}\sum_{a=1}^n k(\mathbf{x}_a,\mathbf{v})k(\mathbf{y}_a,\mathbf{v}') - \mathbb{E}_{\mathbf{xy}}\left[k(\mathbf{x},\mathbf{v})k(\mathbf{y},\mathbf{v}')\right] \right|$$

$$+ J \sup_{\mathbf{v},\mathbf{v}'\in\mathcal{X}} \sup_{k\in\mathcal{K}} \left| \frac{1}{n}\sum_{a=1}^n k(\mathbf{y}_a,\mathbf{v})k(\mathbf{y}_a,\mathbf{v}') - \mathbb{E}_{\mathbf{y}}\left[k(\mathbf{y},\mathbf{v})k(\mathbf{y},\mathbf{v}')\right] \right|$$

by exploiting that $\|\mathbf{A}\|_F = \sup_{\mathbf{B}\in B(1,\mathbf{0})}\langle\mathbf{B},\mathbf{A}\rangle_F$, and $\sum_{i,j=1}^J |B_{ij}| \leq J\|\mathbf{B}\|_F \leq J$ with $\mathbf{B}\in B(1,\mathbf{0})$. Using the bounds obtained for the two terms of Eq. (8), we get

$$\sup_{\mathcal{V}}\sup_{k\in\mathcal{K}} \|\mathbf{S}_n(\mathcal{V}) - \mathbf{\Sigma}(\mathcal{V})\|_F \leq$$

$$\leq \frac{8B^2 J}{n-1} + 2B\sqrt{J}\frac{2n-1}{n-1}\sup_{\mathcal{V}}\sup_{k\in\mathcal{K}}\|\bar{\mathbf{z}}_n - \boldsymbol{\mu}(\mathcal{V})\|_2 + J\left(\|P_n - P\|_{\mathcal{F}_2} + 2\|(P\times Q)_n - (P\times Q)\|_{\mathcal{F}_3} + \|Q_n - Q\|_{\mathcal{F}_2}\right).$$
$$(10)$$

## D.5 Bounding by concentration and the VC property

By combining Eqs. (3), (7) and (10)

$$\sup_{\mathcal{V}}\sup_k \left| \bar{\mathbf{z}}_n^\top(\mathbf{S}_n + \gamma_n I)^{-1}\bar{\mathbf{z}}_n - \boldsymbol{\mu}^\top\mathbf{\Sigma}^{-1}\boldsymbol{\mu} \right| \leq$$

$$\leq \frac{\bar{c}_1}{\gamma_n}\left[ \frac{8B^2 J}{n-1} + 2B\sqrt{J}\frac{2n-1}{n-1}\sqrt{J}\left(\|P_n - P\|_{\mathcal{F}_1} + \|Q_n - Q\|_{\mathcal{F}_1}\right) \right.$$

$$\left. + J\left(\|P_n - P\|_{\mathcal{F}_2} + 2\|(P\times Q)_n - (P\times Q)\|_{\mathcal{F}_3} + \|Q_n - Q\|_{\mathcal{F}_2}\right) \right]$$

$$+ \bar{c}_2\sqrt{J}\left(\|P_n - P\|_{\mathcal{F}_1} + \|Q_n - Q\|_{\mathcal{F}_1}\right) + \bar{c}_3\gamma_n$$

$$= \left(\|P_n - P\|_{\mathcal{F}_1} + \|Q_n - Q\|_{\mathcal{F}_1}\right)\left(\frac{2}{\gamma_n}\bar{c}_1 BJ\frac{2n-1}{n-1} + \bar{c}_2\sqrt{J}\right) + \bar{c}_3\gamma_n$$

$$+ \frac{\bar{c}_1}{\gamma_n}J\left[\|P_n - P\|_{\mathcal{F}_2} + \|Q_n - Q\|_{\mathcal{F}_2} + 2\|(P\times Q)_n - (P\times Q)\|_{\mathcal{F}_3}\right] + \frac{8}{\gamma_n}\frac{\bar{c}_1 B^2 J}{n-1}. \qquad (11)$$

Applying Lemma 3 with $\frac{\delta}{5}$, we get the statement with a union bound. $\qquad\square$

**Lemma 3** (Concentration of the empirical process for uniformly bounded separable Carathéodory VC classes)**.**
*Let $\mathcal{F}$ be*

    *1. VC-subgraph class of $\mathcal{M}\to\mathbb{R}$ functions with VC index $VC(\mathcal{F})$,*

    *2. a uniformly bounded ($\|f\|_{L^\infty(\mathcal{M})}\leq K < \infty, \forall f\in\mathcal{F}$) separable Carathéodory family.*

*Let $\mathbb{Q}$ be a probability measure, and let $\mathbb{Q}_n = \frac{1}{n}\sum_{i=1}^n\delta_{x_i}$ be the corresponding empirical measure. Then for any $\delta\in(0,1)$ with probability at least $1-\delta$*

$$\|\mathbb{Q}_n - \mathbb{Q}\|_{\mathcal{F}} \leq \frac{16\sqrt{2}K}{\sqrt{n}}\left[ 2\sqrt{\log\left[C\times VC(\mathcal{F})(16e)^{VC(\mathcal{F})}\right]} + \frac{\sqrt{2\pi[VC(\mathcal{F})-1]}}{2} \right] + K\sqrt{\frac{2\log\left(\frac{1}{\delta}\right)}{n}}$$

*where the universal constant $C$ is associated according to Lemma 7(iv).*

*Proof.* Notice that $g(x_1,\ldots,x_n) = \|\mathbb{Q}_n - \mathbb{Q}\|_{\mathcal{F}}$ satisfies the bounded difference property with $b = \frac{2K}{n}$ [see Eq. (18)]:

$$\left| g(\mathbf{x}_1,\ldots,\mathbf{x}_n) - g\left(\mathbf{x}_1,\ldots,\mathbf{x}_j,\mathbf{x}_j',\mathbf{x}_{j+1},\ldots,\mathbf{x}_n\right) \right|$$

$$\leq \left| \sup_{f\in\mathcal{F}}\left|\mathbb{Q}f - \frac{1}{n}\sum_{i=1}^n f(\mathbf{x}_i)\right| - \sup_{f\in\mathcal{F}}\left|\mathbb{Q}f - \frac{1}{n}\sum_{i=1}^n f(\mathbf{x}_i) + \frac{1}{n}\left[f(\mathbf{x}_j) - f(\mathbf{x}_j')\right]\right| \right|$$

$$\leq \frac{1}{n}\sup_{f\in\mathcal{F}}|f(\mathbf{x}_j) - f(\mathbf{x}_j')| \leq \frac{1}{n}\left(\sup_{f\in\mathcal{F}}|f(\mathbf{x}_j)| + \sup_{f\in\mathcal{F}}|f(\mathbf{x}_j')|\right) \leq \frac{2K}{n}.$$

Hence, applying Lemma 8, and using symmetrization Steinwart and Christmann (2008) (Prop. 7.10) for the uniformly bounded separable Carathéodory $\mathcal{F}$ class, for arbitrary $\delta \in (0, 1)$ with probability at least $1 - \delta$

$$\|\mathbb{Q}_n - \mathbb{Q}\|_{\mathcal{F}} \leq \mathbb{E}_{x_{1:n}} \|\mathbb{Q}_n - \mathbb{Q}\|_{\mathcal{F}} + K\sqrt{\frac{2\log\left(\frac{1}{\delta}\right)}{n}}$$

$$\leq 2\mathbb{E}_{x_{1:n}} R(\mathcal{F}, x_{1:n}) + K\sqrt{\frac{2\log\left(\frac{1}{\delta}\right)}{n}}.$$

By the Dudley entropy bound Bousquet (2003) [see Eq. (4.4); $\mathrm{diam}(\mathcal{F}, L^2(\mathcal{M}, \mathbb{Q}_n)) \leq 2\sup_{f \in \mathcal{F}} \|f\|_{L^2(\mathcal{M}, \mathbb{Q}_n)} \leq 2\sup_{f \in \mathcal{F}} \|f\|_{L^\infty(\mathcal{M})} \leq 2K < \infty$], Lemma 7(iv) [with $F \equiv K$, $q = 2$ $\mathbb{M} = \mathbb{Q}_n$] and the monotone decreasing property of the covering number, one arrives at

$$R(\mathcal{F}, x_{1:n}) \leq \frac{8\sqrt{2}}{\sqrt{n}} \int_0^{2K} \sqrt{\log N(r, \mathcal{F}, L^2(\mathcal{M}, \mathbb{Q}_n))} \mathrm{d}r$$

$$\leq \frac{8\sqrt{2}}{\sqrt{n}} \left[\int_0^K \sqrt{\log N(r, \mathcal{F}, L^2(\mathcal{M}, \mathbb{Q}_n))} \mathrm{d}r + K\sqrt{\log N(K, \mathcal{F}, L^2(\mathcal{M}, \mathbb{Q}_n))}\right]$$

$$\leq \frac{8\sqrt{2}K}{\sqrt{n}} \left[\int_0^1 \sqrt{\log N(rK, \mathcal{F}, L^2(\mathcal{M}, \mathbb{Q}_n))} \mathrm{d}r + \sqrt{\log N(K, \mathcal{F}, L^2(\mathcal{M}, \mathbb{Q}_n))}\right]$$

$$\leq \frac{8\sqrt{2}K}{\sqrt{n}} \left[\int_0^1 \sqrt{\log\left[a_1\left(\frac{1}{r}\right)^{a_2}\right]} \mathrm{d}r + \sqrt{\log(a_1)}\right] = \frac{8\sqrt{2}K}{\sqrt{n}} \left[2\sqrt{\log(a_1)} + \int_0^1 \sqrt{a_2 \log\left(\frac{1}{r}\right)} \mathrm{d}r\right]$$

$$= \frac{8\sqrt{2}K}{\sqrt{n}} \left[2\sqrt{\log(a_1)} + \frac{\sqrt{\pi a_2}}{2}\right],$$

where $a_1 := C \times VC(\mathcal{F})(16e)^{VC(\mathcal{F})}$, $a_2 := 2[VC(\mathcal{F}) - 1]$ and $\int_0^1 \sqrt{\log\left(\frac{1}{r}\right)} \mathrm{d}r = \int_0^\infty t^{\frac{1}{2}} e^{-t} \mathrm{d}t = \Gamma\left(\frac{3}{2}\right) = \frac{\sqrt{\pi}}{2}$. $\qquad \square$

**Lemma 4** (Properties of $\mathcal{F}_i$ from $\mathcal{K}$)**.**

1. ***Uniform boundedness of $\mathcal{F}_i$-s [see Eqs.*** (1)-(2)***]:*** *If $\mathcal{K}$ is uniformly bounded, i.e., $\exists B < \infty$ such that $\sup_{k \in \mathcal{K}} \sup_{(\mathbf{x}, \mathbf{y}) \in \mathcal{X}^2} |k(\mathbf{x}, \mathbf{y})| \leq B$; then $\mathcal{F}_1$, $\mathcal{F}_2$ and $\mathcal{F}_3$ [Eqs.* (1)-(2)*] are also uniformly bounded with $B$, $B^2$, $B^2$ constants, respectively. That is, $\sup_{k \in \mathcal{K}, \mathbf{v} \in \mathcal{X}} |k(\mathbf{x}, \mathbf{v})| \leq B$, $\sup_{k \in \mathcal{K}, (\mathbf{v}, \mathbf{v}') \in \mathcal{X}^2} |k(\mathbf{x}, \mathbf{v})k(\mathbf{x}, \mathbf{v}')| \leq B^2$, $\sup_{k \in \mathcal{K}, (\mathbf{v}, \mathbf{v}') \in \mathcal{X}^2} |k(\mathbf{x}, \mathbf{v})k(\mathbf{y}, \mathbf{v}')| \leq B^2$.*

2. ***Separability of $\mathcal{F}_i$:*** *since $\mathcal{F}_1$, $\mathcal{F}_2$ and $\mathcal{F}_3$ is parameterized by $\Theta = \mathcal{K} \times \mathcal{X}$, $\mathcal{K} \times \mathcal{X}^2$, $\mathcal{K} \times \mathcal{X}^2$, separability of $\mathcal{K}$ implies that of $\Theta$.*

3. ***Measurability of $\mathcal{F}_i$:*** *$\forall k \in \mathcal{K}$ is measurable, then the elements of $\mathcal{F}_i$ ($i = 1, 2, 3$) are also measurable.* $\square$

## E   Example kernel families

Below we give examples for $\mathcal{K}$ kernel classes for which the associated $\mathcal{F}_i$-s are VC-subgraph and uniformly bounded separable Carathéodory families. The VC property will be a direct consequence of the VC indices of finite-dimensional function classes and preservation theorems (see Lemma 7); for a nice example application see Srebro and Ben-David (2006) (Section 5) who study the pseudo-dimension of $(\mathbf{x}, \mathbf{y}) \mapsto k(\mathbf{x}, \mathbf{y})$ kernel classes, for different Gaussian families. We take these Gaussian classes (isotropic, full) and use the preservation trick to bound the VC indices of the associated $\mathcal{F}_i$-s.

**Lemma 5** ($\mathcal{F}_i$-s are VC-subgraph and uniformly bounded separable Carathéodory families for isotropic Gaussian kernel)**.** *Let $\mathcal{K} = \left\{k_\sigma : (\mathbf{x}, \mathbf{y}) \in \mathcal{X} \times \mathcal{X} \subseteq \mathbb{R}^d \times \mathbb{R}^d \mapsto e^{-\frac{\|\mathbf{x} - \mathbf{y}\|_2^2}{2\sigma^2}} : \sigma > 0\right\}$. Then the $\mathcal{F}_1$, $\mathcal{F}_2$, $\mathcal{F}_3$ classes [see Eqs.* (1)-(2)*] associated to $\mathcal{K}$ are*

- *VC-subgraphs with indices $VC(\mathcal{F}_1) \leq d + 4$, $VC(\mathcal{F}_2) \leq d + 4$, $VC(\mathcal{F}_3) \leq 2d + 4$, and*

- *uniformly bounded separable Carathéodory families, with $\|f\|_{L^\infty(\mathcal{M})} \leq 1$ for all $f \in \{\mathcal{F}_1, \mathcal{F}_2, \mathcal{F}_3\}$.[4]*

*Proof.* **VC subgraph property:**

- $\mathcal{F}_1$: Consider the function class $\mathcal{G} = \left\{ \mathbf{x} \mapsto \frac{\|\mathbf{x}-\mathbf{v}\|_2^2}{2\sigma^2} = \frac{1}{2\sigma^2}\left(\|\mathbf{x}\|_2^2 - 2\langle \mathbf{x}, \mathbf{v}\rangle_2 + \|\mathbf{v}\|_2^2\right) : \sigma > 0, \mathbf{v} \in \mathcal{X} \right\} \subseteq$
  $L^0(\mathbb{R}^d)$. $\mathcal{G} \subseteq \tilde{\mathcal{G}} := span\left(\mathbf{x} \mapsto \|\mathbf{x}\|_2^2, \{\mathbf{x} \mapsto x_i\}_{i=1}^d, \mathbf{x} \mapsto 1\right)$ vector space, $dim(\mathcal{G}) \leq d+2$. Thus
  by Lemma 7(i)-(ii), $\mathcal{G}$ is VC with $VC(\mathcal{G}) \leq d+4$; applying Lemma 7(iii) with $\phi(z) = e^{-z}$, $\mathcal{F}_1 = \phi \circ \mathcal{G}$
  is also VC with index $VC(\mathcal{F}_1) \leq d+4$.

- $\mathcal{F}_2$: Since $\mathcal{F}_2 = \left\{ \mathbf{x} \mapsto k(\mathbf{x}, \mathbf{v})k(\mathbf{x}, \mathbf{v}') = e^{-\frac{\|\mathbf{x}-\mathbf{v}\|_2^2 + \|\mathbf{x}-\mathbf{v}'\|_2^2}{2\sigma^2}} : \sigma > 0, \mathbf{v} \in \mathcal{X}, \mathbf{v}' \in \mathcal{X} \right\}$,
  and $\left\{ \mathbf{x} \mapsto \frac{\|\mathbf{x}-\mathbf{v}\|_2^2 + \|\mathbf{x}-\mathbf{v}'\|_2^2}{2\sigma^2} : \sigma > 0, \mathbf{v} \in \mathcal{X}, \mathbf{v}' \in \mathcal{X} \right\} \subseteq S =$
  $span\left(\mathbf{x} \mapsto \|\mathbf{x}\|_2^2, \{\mathbf{x} \mapsto x_i\}_{i=1}^d, \mathbf{x} \mapsto 1\right), VC(\mathcal{F}_2) \leq d+4$.

- $\mathcal{F}_3$: Since

$$\mathcal{F}_3 = \left\{ (\mathbf{x}, \mathbf{y}) \mapsto k(\mathbf{x}, \mathbf{v})k(\mathbf{y}, \mathbf{v}') = e^{-\frac{\|\mathbf{x}-\mathbf{v}\|^2 + \|\mathbf{y}-\mathbf{v}'\|_2^2}{2\sigma^2}} = e^{-\frac{\|[\mathbf{x};\mathbf{y}]-[\mathbf{v};\mathbf{v}']\|_2^2}{2\sigma^2}} : \sigma > 0, \mathbf{v} \in \mathbb{R}^d, \mathbf{v}' \in \mathbb{R}^d \right\},$$

  from the result on $\mathcal{F}_1$ we get that $VC(\mathcal{F}_3) \leq 2d+4$.

**Uniformly bounded, separable Carathéodory family:**

The result follows from Lemma 4 by noting that $|k(\mathbf{x}, \mathbf{y})| \leq 1 =: B$, $(\mathbf{x}, \mathbf{y}) \mapsto e^{-\frac{\|\mathbf{x}-\mathbf{y}\|_2^2}{2\sigma^2}}$ is continuous
$(\forall \sigma > 0)$, $\mathbb{R}^+$ is separable, and the $(\sigma, \mathbf{v}) \mapsto e^{-\frac{\|\mathbf{x}-\mathbf{v}\|_2^2}{2\sigma^2}}$, $(\sigma, \mathbf{v}, \mathbf{v}') \mapsto e^{-\frac{\|\mathbf{x}-\mathbf{v}\|_2^2}{2\sigma^2}}e^{-\frac{\|\mathbf{x}-\mathbf{v}'\|_2^2}{2\sigma^2}}$, $(\sigma, \mathbf{v}, \mathbf{v}') \mapsto$
$e^{-\frac{\|\mathbf{x}-\mathbf{v}\|_2^2}{2\sigma^2}}e^{-\frac{\|\mathbf{y}-\mathbf{v}'\|_2^2}{2\sigma^2}}$ mappings are continuous $(\forall \mathbf{x}, \mathbf{y} \in \mathcal{X})$. $\square$

**Lemma 6** ($\mathcal{F}_i$-s are VC-subgraph and uniformly bounded separable Carathéodory families for full Gaussian
kernel). *Let $\mathcal{K} = \{k_{\mathbf{A}} : (\mathbf{x}, \mathbf{y}) \in \mathcal{X} \times \mathcal{X} \subseteq \mathbb{R}^d \times \mathbb{R}^d \mapsto e^{-(\mathbf{x}-\mathbf{y})^\top \mathbf{A}(\mathbf{x}-\mathbf{y})} : \mathbf{A} \succeq \mathbf{0}\}$. Then the $\mathcal{F}_1$, $\mathcal{F}_2$, $\mathcal{F}_3$
classes [see Eqs. (1)-(2)] associated to $\mathcal{K}$ are*

- *VC-subgraphs with indices $VC(\mathcal{F}_1) \leq \frac{d(d+1)}{2} + d + 2$, $VC(\mathcal{F}_2) \leq \frac{d(d+1)+2}{2} + d + 2$, $VC(\mathcal{F}_3) \leq d(d+1) + 2d + 3$,*

- *uniformly bounded separable Carathéodory families, with $\|f\|_{L^\infty(\mathcal{M})} \leq 1$ for all $f \in \{\mathcal{F}_1, \mathcal{F}_2, \mathcal{F}_3\}$.[4]*

*Proof.* We prove the VC index values; the rest is essentially identical to the proof of Lemma 5.

- $\mathcal{F}_1$: Using that $\mathcal{G} = \left\{\mathbf{x} \mapsto (\mathbf{x}-\mathbf{v})^\top \mathbf{A}(\mathbf{x}-\mathbf{v}) : \mathbf{A} \succeq \mathbf{0}, \mathbf{v} \in \mathcal{X}\right\} \subseteq S :=$
  $span\left(\{\mathbf{x} \mapsto x_i x_j\}_{1 \leq i \leq j \leq d}, \{\mathbf{x} \mapsto x_i\}_{1 \leq i \leq d}, \mathbf{x} \mapsto 1\right)$, we have $VC(\mathcal{F}_1) \leq VC(\mathcal{G}) \leq dim(S) + 2 \leq$
  $\frac{d(d+1)}{2} + d + 3$.

- $\mathcal{F}_2$: Since $\mathcal{F}_2 = \left\{\mathbf{x} \mapsto k(\mathbf{x}, \mathbf{v})k(\mathbf{x}, \mathbf{v}') = e^{-\left[(\mathbf{x}-\mathbf{v})^\top \mathbf{A}(\mathbf{x}-\mathbf{v})+(\mathbf{x}-\mathbf{v}')^\top \mathbf{A}(\mathbf{x}-\mathbf{v}')\right]} : \mathbf{A} \succeq \mathbf{0}, \mathbf{v} \in \mathcal{X}, \mathbf{v}' \in \mathcal{X}\right\}$,
  and

$$\left\{(\mathbf{x}, \mathbf{y}) \mapsto (\mathbf{x}-\mathbf{v})^\top \mathbf{A}(\mathbf{x}-\mathbf{v}) + (\mathbf{x}-\mathbf{v}')^\top \mathbf{A}(\mathbf{x}-\mathbf{v}')\right\} \subseteq S$$
$$:= span\left(\{\mathbf{x} \mapsto x_i x_j\}_{1 \leq i \leq j \leq d}, \{\mathbf{x} \mapsto x_i\}_{1 \leq i \leq d}, \mathbf{x} \mapsto 1\right),$$

  we have $VC(\mathcal{F}_2) \leq VC(S) = dim(S) + 2 \leq \frac{d(d+1)}{2} + d + 3$.

- $\mathcal{F}_3$: Exploiting that

$$\mathcal{F}_3 = \left\{ (\mathbf{x}, \mathbf{y}) \mapsto k(\mathbf{x}, \mathbf{v})k(\mathbf{y}, \mathbf{v}') = e^{-\left[(\mathbf{x}-\mathbf{v})^\top \mathbf{A}(\mathbf{x}-\mathbf{v})+(\mathbf{y}-\mathbf{v}')^\top \mathbf{B}(\mathbf{y}-\mathbf{v}')\right]} : \mathbf{A} \succeq 0, \mathbf{B} \succeq 0, \mathbf{v} \in \mathcal{X}, \mathbf{v}' \in \mathcal{X} \right\},$$

and $\{(\mathbf{x}, \mathbf{y}) \mapsto (\mathbf{x} - \mathbf{v})^\top \mathbf{A}(\mathbf{x} - \mathbf{v}) + (\mathbf{y} - \mathbf{v}')^\top \mathbf{B}(\mathbf{y} - \mathbf{v}')\} \subseteq S :=$ $span\left(\{(\mathbf{x}, \mathbf{y}) \mapsto x_i x_j\}_{1 \le i \le j \le d}, \{(\mathbf{x}, \mathbf{y}) \mapsto x_i\}_{1 \le i \le d}, (\mathbf{x}, \mathbf{y}) \mapsto 1, \{(\mathbf{x}, \mathbf{y}) \mapsto y_i y_j\}_{1 \le i \le j \le d}, \{(\mathbf{x}, \mathbf{y}) \mapsto y_i\}_{1 \le i \le d}\right),$ we have $VC(\mathcal{F}_3) \le VC(S) = dim(S) + 2 \le d(d+1) + 2d + 3.$

$\square$

# F  Proof of proposition 1

Recall Proposition 1:

**Proposition 1** (Lower bound on ME test power)**.** *Let $\mathcal{K}$ be a uniformly bounded (i.e., $\exists B < \infty$ such that $\sup_{k \in \mathcal{K}} \sup_{(\mathbf{x}, \mathbf{y}) \in \mathcal{X}^2} |k(\mathbf{x}, \mathbf{y})| \le B$) family of $k : \mathcal{X} \times \mathcal{X} \to \mathbb{R}$ measurable kernels. Let $\mathbb{V}$ be a collection in which each element is a set of $J$ test locations. Assume that $\tilde{c} := \sup_{\mathcal{V} \in \mathbb{V}, k \in \mathcal{K}} \|\mathbf{\Sigma}^{-1}\|_F < \infty$. For large $n$, the test power $\mathbb{P}\left(\hat{\lambda}_n \ge T_\alpha\right)$ of the ME test satisfies $\mathbb{P}\left(\hat{\lambda}_n \ge T_\alpha\right) \ge L(\lambda_n)$ where*

$$L(\lambda_n) := 1 - 2e^{-\xi_1(\lambda_n - T_\alpha)^2/n} - 2e^{-\frac{[\gamma_n(\lambda_n - T_\alpha)(n-1) - \xi_2 n]^2}{\xi_3 n(2n-1)^2}} - 2e^{-[(\lambda_n - T_\alpha)/3 - \bar{c}_3 n \gamma_n]^2 \gamma_n^2 / \xi_4},$$

*and $\bar{c}_3, \xi_1, \dots \xi_4$ are positive constants depending on only $B, J$ and $\tilde{c}$. The parameter $\lambda_n := n\boldsymbol{\mu}^\top \mathbf{\Sigma}^{-1} \boldsymbol{\mu}$ is the population counterpart of $\hat{\lambda}_n := n\bar{\mathbf{z}}_n^\top \left(\mathbf{S}_n + \gamma_n I\right)^{-1} \bar{\mathbf{z}}_n$ where $\boldsymbol{\mu} = \mathbb{E}_{\mathbf{xy}}[\mathbf{z}_1]$ and $\mathbf{\Sigma} = \mathbb{E}_{\mathbf{xy}}[(\mathbf{z}_1 - \boldsymbol{\mu})(\mathbf{z}_1 - \boldsymbol{\mu})^\top]$. For large $n$, $L(\lambda_n)$ is increasing in $\lambda_n$.*

## F.1  Proof

By (3), we have

$$|\hat{\lambda}_n - \lambda_n| \le \frac{\bar{c}_1 n}{\gamma_n} \|\mathbf{\Sigma} - \mathbf{S}_n\|_F + \bar{c}_2 n \|\bar{\mathbf{z}}_n - \boldsymbol{\mu}\|_2 + \bar{c}_3 n \gamma_n. \tag{12}$$

We will bound each of the three terms in (12).

**Bounding $\|\bar{\mathbf{z}}_n - \boldsymbol{\mu}\|_2$ (second term in (12))**

Let $g(\mathbf{x}, \mathbf{y}, \mathbf{v}) := k(\mathbf{x}, \mathbf{v}) - k(\mathbf{y}, \mathbf{v})$. Define $\mathbf{v}^* := \arg \max_{\mathbf{v} \in \{\mathbf{v}_1, \dots, \mathbf{v}_J\}} \left| \frac{1}{n} \sum_{i=1}^n g(\mathbf{x}_i, \mathbf{y}_i, \mathbf{v}) - \mathbb{E}_{\mathbf{xy}}[g(\mathbf{x}, \mathbf{y}, \mathbf{v})] \right|$. Define $G_i := g(\mathbf{x}_i, \mathbf{y}_i, \mathbf{v}^*)$.

$$\begin{aligned}
\|\bar{\mathbf{z}}_n - \boldsymbol{\mu}\|_2 &= \sup_{\mathbf{b} \in B(1,\mathbf{0})} \langle \mathbf{b}, \bar{\mathbf{z}}_n - \boldsymbol{\mu} \rangle_2 \\
&\le \sup_{\mathbf{b} \in B(1,\mathbf{0})} \sum_{j=1}^J |b_j| \left| \frac{1}{n} \sum_{i=1}^n [k(\mathbf{x}_i, \mathbf{v}_j) - k(\mathbf{y}_i, \mathbf{v}_j)] - \mathbb{E}_{\mathbf{xy}}[k(\mathbf{x}, \mathbf{v}_j) - k(\mathbf{y}, \mathbf{v}_j)] \right| \\
&= \sup_{\mathbf{b} \in B(1,\mathbf{0})} \sum_{j=1}^J |b_j| \left| \frac{1}{n} \sum_{i=1}^n g(\mathbf{x}_i, \mathbf{y}_i, \mathbf{v}_j) - \mathbb{E}_{\mathbf{xy}}[g(\mathbf{x}, \mathbf{y}, \mathbf{v}_j)] \right| \\
&\le \left| \frac{1}{n} \sum_{i=1}^n G_i - \mathbb{E}_{\mathbf{xy}}[G_1] \right| \sup_{\mathbf{b} \in B(1,\mathbf{0})} \sum_{j=1}^J |b_j| \\
&\le \sqrt{J} \left| \frac{1}{n} \sum_{i=1}^n G_i - \mathbb{E}_{\mathbf{xy}}[G_1] \right| \sup_{\mathbf{b} \in B(1,\mathbf{0})} \|\mathbf{b}\|_2 \\
&= \sqrt{J} \left| \frac{1}{n} \sum_{i=1}^n G_i - \mathbb{E}_{\mathbf{xy}}[G_1] \right|,
\end{aligned}$$

where we used the fact that $\|\mathbf{b}\|_1 \le \sqrt{J}\|\mathbf{b}\|_2$. It can be seen that $-2B \le G_i \le 2B$ because

$$G_i = k(\mathbf{x}_i, \mathbf{v}^*) - k(\mathbf{y}_i, \mathbf{v}^*) \le |k(\mathbf{x}_i, \mathbf{v}^*)| + |k(\mathbf{y}_i, \mathbf{v}^*)| \le 2B.$$

Using Hoeffding's inequality (Lemma 9) to bound $\left|\frac{1}{n}\sum_{i=1}^n G_i - \mathbb{E}_{\mathbf{xy}}[G_1]\right|$, we have

$$\mathbb{P}\left(n\bar{c}_2\|\bar{\mathbf{z}}_n - \boldsymbol{\mu}\|_2 \le \alpha\right) \ge 1 - 2\exp\left(-\frac{\alpha^2}{8B^2\bar{c}_2^2 Jn}\right). \tag{13}$$

**Bounding first ($\|\boldsymbol{\Sigma} - \mathbf{S}_n\|_F$) and third terms in (12)**

Let $\eta(\mathbf{v}_i, \mathbf{v}_j) := \left|\frac{1}{n}\sum_{a=1}^n g(\mathbf{x}_a, \mathbf{y}_a, \mathbf{v}_i)g(\mathbf{x}_a, \mathbf{y}_a, \mathbf{v}_j) - \mathbb{E}_{\mathbf{xy}}[g(\mathbf{x}, \mathbf{y}, \mathbf{v}_i)g(\mathbf{x}, \mathbf{y}, \mathbf{v}_j)]\right|$. Define $(\mathbf{v}_1^*, \mathbf{v}_2^*) = \arg\max_{(\mathbf{v}^{(1)}, \mathbf{v}^{(2)}) \in \{(\mathbf{v}_i, \mathbf{v}_j)\}_{i,j=1}^J} \eta(\mathbf{v}^{(1)}, \mathbf{v}^{(2)})$. Define $H_i := g(\mathbf{x}_i, \mathbf{y}_i, \mathbf{v}_1^*)g(\mathbf{x}_i, \mathbf{y}_i, \mathbf{v}_2^*)$. By (8), we have

$$\|\mathbf{S}_n - \boldsymbol{\Sigma}\|_F \le (*_1) + (*_2),$$

$$(*_1) = \left\|\frac{1}{n}\sum_{a=1}^n \mathbf{z}_a\mathbf{z}_a^\top - \mathbb{E}_{\mathbf{xy}}[\mathbf{z}_1\mathbf{z}_1^\top]\right\|_F,$$

$$(*_2) = \frac{8B^2 J}{n-1} + 2B_k\sqrt{J}\frac{2n-1}{n-1}\|\bar{\mathbf{z}}_n - \boldsymbol{\mu}\|_2.$$

We can upper bound $(*_2)$ by applying Hoeffding's inequality to bound $\|\bar{\mathbf{z}}_n - \boldsymbol{\mu}\|_2$ giving

$$\mathbb{P}\left(\frac{\bar{c}_1 n}{\gamma_n}(*_2) \le \alpha\right) \ge 1 - 2\exp\left(-\frac{(\alpha\gamma_n - \alpha\gamma_n n + 8B^2\bar{c}_1 Jn)^2}{32B^4\bar{c}_1^2 J^2 n(2n-1)^2}\right). \tag{14}$$

We can upper bound $(*_1)$ with

$$(*_1) = \sup_{\mathbf{B} \in B(1,\mathbf{0})} \left\langle \mathbf{B}, \frac{1}{n}\sum_{a=1}^n \mathbf{z}_a\mathbf{z}_a^\top - \mathbb{E}_{\mathbf{xy}}[\mathbf{z}_1\mathbf{z}_1^\top]\right\rangle_F$$

$$\le \sup_{\mathbf{B} \in B(1,\mathbf{0})} \sum_{i=1}^J\sum_{j=1}^J |B_{ij}| \left|\frac{1}{n}\sum_{a=1}^n g(\mathbf{x}_a, \mathbf{y}_a, \mathbf{v}_i)g(\mathbf{x}_a, \mathbf{y}_a, \mathbf{v}_j) - \mathbb{E}_{\mathbf{xy}}[g(\mathbf{x}, \mathbf{y}, \mathbf{v}_i)g(\mathbf{x}, \mathbf{y}, \mathbf{v}_j)]\right|$$

$$\le \left|\frac{1}{n}\sum_{a=1}^n H_a - \mathbb{E}_{\mathbf{xy}}[H_1]\right| \sup_{\mathbf{B} \in B(1,\mathbf{0})} \sum_{i=1}^J\sum_{j=1}^J |B_{ij}|$$

$$\le J\left|\frac{1}{n}\sum_{a=1}^n H_a - \mathbb{E}_{\mathbf{xy}}[H_1]\right| \sup_{\mathbf{B} \in B(1,\mathbf{0})} \|\mathbf{B}\|_F = J\left|\frac{1}{n}\sum_{a=1}^n H_a - \mathbb{E}_{\mathbf{xy}}[H_1]\right|,$$

where we used the fact that $\sum_{i=1}^J\sum_{j=1}^J |B_{ij}| \le J\|\mathbf{B}\|_F$. It can be seen that $-4B^2 \le H_a \le 4B^2$. Using Hoeffding's inequality (Lemma 9) to bound $\left|\frac{1}{n}\sum_{a=1}^n H_a - \mathbb{E}_{\mathbf{xy}}[H_1]\right|$, we have

$$\mathbb{P}\left(\frac{\bar{c}_1 n}{\gamma_n}(*_1) \le \alpha\right) \ge 1 - 2\exp\left(-\frac{\alpha^2\gamma_n^2}{32B^4 J^2\bar{c}_1^2 n}\right), \tag{15}$$

implying that

$$\mathbb{P}\left(\frac{\bar{c}_1 n}{\gamma_n}(*_1) + \bar{c}_3 n\gamma_n \le \alpha\right) \ge 1 - 2\exp\left(-\frac{(\alpha - \bar{c}_3 n\gamma_n)^2\gamma_n^2}{32B^4 J^2\bar{c}_1^2 n}\right). \tag{16}$$

Applying a union bound on (13), (14), and (16) with $t = \alpha/3$, we can conclude that

$$\mathbb{P}\left(\left|\hat{\lambda}_n - \lambda_n\right| \le t\right) \ge \mathbb{P}\left(\frac{\bar{c}_1 n}{\gamma_n}\|\boldsymbol{\Sigma} - \mathbf{S}_n\|_F + \bar{c}_2 n\|\bar{\mathbf{z}}_n - \boldsymbol{\mu}\|_2 + \bar{c}_3 n\gamma_n \le t\right)$$

$$\ge 1 - 2\exp\left(-\frac{t^2}{3^2 \cdot 8B^2\bar{c}_2^2 Jn}\right) - 2\exp\left(-\frac{(t\gamma_n n - t\gamma_n - 24B^2\bar{c}_1 Jn)^2}{3^2 \cdot 32B^4\bar{c}_1^2 J^2 n(2n-1)^2}\right) - 2\exp\left(-\frac{(t/3 - \bar{c}_3 n\gamma_n)^2\gamma_n^2}{32B^4 J^2\bar{c}_1^2 n}\right).$$

A rearrangement yields

$$\mathbb{P}\left(\hat{\lambda}_n \ge T_\alpha\right)$$

$$\geq 1 - 2 \exp\left(-\frac{(\lambda_n - T_\alpha)^2}{3^2 \cdot 8 B^2 \overline{c}_2^2 J n}\right) - 2\exp\left(-\frac{(\gamma_n(\lambda_n - T_\alpha)(n-1) - 24 B^2 \overline{c}_1 J n)^2}{3^2 \cdot 32 B^4 \overline{c}_1^2 J^2 n (2n-1)^2}\right) - 2\exp\left(-\frac{((\lambda_n - T_\alpha)/3 - \overline{c}_3 n \gamma_n)^2 \gamma_n^2}{32 B^4 J^2 \overline{c}_1^2 n}\right).$$

Define $\xi_1 := \frac{1}{3^2 \cdot 8 B^2 \overline{c}_2^2 J}, \xi_2 := 24 B^2 \overline{c}_1 J, \xi_3 := 3^2 \cdot 32 B^4 \overline{c}_1^2 J^2, \xi_4 := 32 B^4 J^2 \overline{c}_1^2$. We have

$$\mathbb{P}\left(\hat{\lambda}_n \geq T_\alpha\right)$$

$$\geq 1 - 2\exp\left(-\frac{\xi_1(\lambda_n - T_\alpha)^2}{n}\right) - 2\exp\left(-\frac{(\gamma_n(\lambda_n - T_\alpha)(n-1) - \xi_2 n)^2}{\xi_3 n(2n-1)^2}\right) - 2\exp\left(-\frac{((\lambda_n - T_\alpha)/3 - \overline{c}_3 n \gamma_n)^2 \gamma_n^2}{\xi_4}\right).$$

$\square$

# G   External lemmas

In this section we detail some external lemmas used in our proof.

**Lemma 7** (properties of VC classes, see page 141, 146-147 in van der Vaart and Wellner (2000) and page 160-161 in Kosorok (2008))**.**

  (i) *Monotonicity:* $\mathcal{G} \subseteq \tilde{\mathcal{G}} \subseteq L^0(\mathcal{M}) \Rightarrow VC(\mathcal{G}) \leq VC(\tilde{\mathcal{G}})$.

  (ii) *Finite-dimensional vector space: if $\mathcal{G}$ is a finite-dimensional vector space of measurable functions, then* $VC(\mathcal{G}) \leq dim(\mathcal{G}) + 2$.

  (iii) *Composition with monotone function: If $\mathcal{G}$ is VC and $\phi : \mathbb{R} \to \mathbb{R}$ is monotone, then for $\phi \circ \mathcal{G} := \{\phi \circ g : g \in \mathcal{G}\}, VC(\phi \circ \mathcal{G}) \leq VC(\mathcal{G})$.*

  (iv) *The $r$-covering number of a VC class grows only polynomially in $\frac{1}{r}$: Let $\mathcal{F}$ be VC on the domain $\mathcal{M}$ with measurable envelope $F$ ($|f(m)| \leq F(m), \forall m \in \mathcal{M}, f \in \mathcal{F}$). Then for any $q \geq 1$ and $\mathbb{M}$ probability measure for which $\|F\|_{L^q(\mathcal{M}, \mathbb{M})} > 0$*

$$N\left(r\|F\|_{L^q(\mathcal{M}, \mathbb{M})}, \mathcal{F}, L^q(\mathcal{M}, \mathbb{M})\right) \leq C \times VC(\mathcal{F})(16e)^{VC(\mathcal{F})}\left(\frac{1}{r}\right)^{q[VC(\mathcal{F})-1]} \tag{17}$$

  *for any $r \in (0,1)$ with a universal constant $C$.*

**Lemma 8** (McDiarmid's inequality)**.** *Let $X_1, \ldots, X_n \in \mathcal{M}$ be independent random variables and let $g : \mathcal{M}^n \to \mathbb{R}$ be a function such that the*

$$\sup_{\mathbf{x}_1, \ldots, \mathbf{x}_n, \mathbf{x}_j' \in \mathcal{M}} \left|g(\mathbf{x}_1, \ldots, \mathbf{x}_n) - g\left(\mathbf{x}_1, \ldots, \mathbf{x}_j, \mathbf{x}_j', \mathbf{x}_{j+1}, \ldots, \mathbf{x}_n\right)\right| \leq b \tag{18}$$

*bounded difference property holds. Then for arbitrary $\delta \in (0,1)$*

$$\mathbb{P}\left(g(X_1, \ldots, X_n) \leq \mathbb{E}[g(X_1, \ldots, X_n)] + b\sqrt{\frac{\log\left(\frac{1}{\delta}\right)n}{2}}\right) \geq 1 - \delta.$$

**Lemma 9** (Hoeffding's inequality)**.** *Let $X_1, \ldots, X_n$ be i.i.d. random variables with $\mathbb{P}(a \leq X_i \leq b) = 1$. Let $\overline{X} := \frac{1}{n}\sum_{i=1}^n X_i$. Then,*

$$\mathbb{P}\left(\left|\overline{X} - \mathbb{E}[\overline{X}]\right| \leq t\right) \geq 1 - 2\exp\left(-\frac{2nt^2}{(b-a)^2}\right).$$

*Equivalently, for any $\delta \in (0,1)$, with probability at least $1 - \delta$, it holds that*

$$\left|\overline{X} - \mathbb{E}[\overline{X}]\right| \leq \frac{b-a}{\sqrt{2n}}\sqrt{\log(2/\delta)}.$$

## Footnotes

[2]`pdftotext` is available at `http://poppler.freedesktop.org`.

[3]WordNet is available online at `https://wordnet.princeton.edu/wordnet/citing-wordnet/`.

[4] $\mathcal{M} = \mathcal{X}$ for $\mathcal{F}_1$ and $\mathcal{F}_2$, and $\mathcal{M} = \mathcal{X}^2$ in case of $\mathcal{F}_3$.