[Reviews · NeurIPS 2016]

Reviewer 1

Summary

The paper deals with the problem of detecting features from which two probability distributions may be distinguished at best, when they are only observed through finite i.i.d. samples. The approach proposed here starts from a lower bound on the power of the ME and SCF two-sample tests based on semi-metrics which are defined by Chwialkowsky et al. (2015) as the sum of differences of expectations of analytic functions evaluated at spatial or frequency locations (the features). As this lower bound is increasing with respect to some quantity that can be estimated from a held-out data set, the idea is to maximize this estimate with respect to the kernel parameters but also with respect to the features, which enables to extract a set of interpretable features designed to best distinguish the two probability distributions.

Qualitative Assessment

The present work provides significant improvements to the paper by Chwialkowsky et al. (2015), as it at least partially answers the crucial question of the kernel parameters and features choice in ME and SCF two-sample tests based on differences of expectations of analytic functions computed at particular features, derived from the now classical MMD tests. The ideas it develops are very interesting and promising, and I am sure they will lead to many new insights in two-sample tests and relative issues. This work however still lets unsolved or raises important questions, that should be underlined. First notice that the approach is only valid when the chosen critical values of the tests (so that the test has the desired significance level) do not depend on the kernel parameters, which is the case here only because the critical values are chosen as quantiles of the asymptotic chi-square distribution. As far as I understand it, it could not be used with permutation or bootstrap based critical values often considered in the original MMD tests. Then notice that the choice of the number of features remains an unsolved issue. Indeed, in experiments the number of features seems to be arbitrarily fixed very small. Is this choice optimal? To answer this question would imply deeply studying the power of the final test, to see whether this choice is optimal or not. Of course, this is closely linked to the runtime efficiency, and this leads me to my last main comment. A precise evaluation of the algorithmic complexity of the proposed approach would be appreciable (including the optimization gradient ascent algorithm), in particular when comparisons with the original MMD or ME and SCF tests are produced. I now list below my less major comments. I found the paper rather pleasant to read, but maybe difficult to understand for nonspecialists of two-sample tests, or without studying before the papers by Gretton et al. (2012), Zaremba et al. (2013), and Chwialkowsky et al. (2015). Please try to make the reading more fluent for nonspecialists. Could the abstract be rewritten to fit this comment ? Could the proposed approach be summarized in an algorithmic form to make it more easily understandable (including the optimization gradient ascent algorithm) ? Could a final bound on the final test (including the optimization process) be precisely stated at the end of Section 3 ? A conclusion/discussion section should also be added. The experiments are interesting, but not detailed enough especially for the articles and images applications (the supplementary material can be completed). The choices made for the kernel, the number of features, etc. should be more clearly highlighted and discussed. Minor remarks: please add "when possible" after "the null distribution" line 80 page 2: the null distribution is not known in the most general two-sample problems. Otherwise add "asymptotic" in front of "distribution" and "null distribution"; in Proposition 1, $\Sigma$ is not defined; the last paragraph on page 3 has to be rewritten: to my mind, it is a bit far-fetched and contains some mistakes. The notation $O(.)$ should be consistent pages 5 and 7.

Confidence in this Review

2-Confident (read it all; understood it all reasonably well)


Reviewer 2

Summary

The paper considers the classical two sample problem where one wants, from the observation of two independent samples in R^d, X_1^n and Y_1^n with respective distribution P and Q, be able to test whether P = Q or P \neq Q. The solution investigated in the paper, which is due to Chwialkowski et al., is very interesting. It is based on a kernel k : (R^d)^2 → R and a set of J test locations (v_j)^J_{j=1}. For any i = 1,...,n, one builds the vectors z_i with coordinates k(x_i,v_j) − k(y_i,v_j), j = 1,...,J and measure the discrepancy between P and Q with the T-statistic \hat{\lambda}_n=n\bar{z}_n(S_n+\gamma_nI)^{-1}\bar{z}_n, where \bar{z}_n=\frac1n\sum_{i=1}^nz_i, S_n=\frac1{n-1}\sum_{i=1}^n(z_i-\bar{z}_n)(z_i-\bar{z}_n)^T. The authors Chwialkowski et al. proved in particular that the asymptotic distribution of \hat{\lambda}_n, when n → ∞ and the other parameters are fixed, is \chi^2(J). The present paper also consider an associated test, called SCF-test which relies on the same ideas. The main question is how should we choose the test locations (vj)_{j=1}^J to optimize the performance of this test (the choice of the kernel k or of some scaling parameters is also relevant). This is precisely the question considered by the author, who proceed as follows. First, they prove a lower bound on the power of the test that is increasing with the limit \lambda = \mu^T \Sigma^{−1}\mu of \hat{\lambda}_n/n as n → ∞. Then, as \lambda is unknown, they cut the sample in two halves, the first one is used to estimate the parameters \mu and \Sigma and therefore estimate the function \lambda of the parameters of interest (v_j)_{j=1}^J. The second is used to select the parameters maximizing this estimator of \lambda. The author proves the consistency of the estimator of \lambda and test its procedure on both toy Gaussian regression models and in two real data-sets in a simulation study concluding the paper.

Qualitative Assessment

I found the approach of the problem very interesting and the solution quite elegant and, even if I did not check the details of all proofs, they seem correct. This is why I would like to see the paper published. However, and this is my main concern, I really think that the author should make a strong effort of presentation of their results. In the present state, I think that it is quite hard to read due to the lack of many details, not completely honest on some aspects and both theorems are unnecessarily way too technical. I will now detail some of these points. You should improve the presentation of the paper. Some important formulas, like the definition of your main quantities (for example zi p3 l98) should be isolated in an equation to ease the reading. Moreover, please define the quantities in the text when you use them in your main results, for example the Frobenius norm used p3 l121 in your first result is only defined in Section C1 of the supplementary material. These are really not details, as I said it makes the paper very hard to understand. To improve the presentation, I also suggest that you avoid the sketchs of proofs that I found quite obscur and not really informative. The same holds for the comment after Proposition 1 giving another intuition that maximizing \lambda would lead to better performances, as you said, Proposition 1 does it and better. Also, p4 at the beginning of the Section Contribution, you briefly change from a parameter k to designate the kernel to \sigma before going back to k and use \sigma as a tuning parameter. You refer several times to the fact that your procedure is adapted to the large dimension setting. You should be more precise here. I believed that it meant that d or J can be large compared to the number n of observations. However, it looks like your test has only confidence level \alpha in an asymptotic framework where n → ∞ while the other parameters remain fixed? Also, in your comment after Theorem 2, you get the rate O(n^{−1/4}) only if the VC dimension of your class of kernels is not allowed to grow with n. Finally, in your simulations study, I think that, in all toy models, n is larger than d and much larger than J, and, for example on Fig 3, it is not clear that your procedure remain effective when d gets closer to n. You should finally improve the presentation of your theorems. For example, since you introduce some ”constants” in Proposition 1 depending on B, J to express L(\lambda_n), why do these parameters and meaningless absolute constants still appear in the expression of L(\lambda_n)? You should really write a more synthetic and comprehensive result in the main text, and probably keep this result for the supplementary material. The same comments hold for Theorem 2.

Confidence in this Review

2-Confident (read it all; understood it all reasonably well)


Reviewer 3

Summary

The authors propose to optimize the parameters of a two-sample test for the equality of two distributions with kernel and Fourier features. The The tunable parameters are the width of a Gaussian kernel and the coordinates of test locations. A lower bound on the test power is derived (which then acts as an objective function), and consistency is established. The discriminative power of different variants of the test is empirically compared to MMD-based tests. As a nice side product, the optimized test locations can yield easy-to-interpret discrimination functions.

Qualitative Assessment

The overall approach is convincing, and the results look very good. However, the paper has a few potentially weak spots on which I would like to get comments from the authors. 1.) How do you ensure the assumption that $\tilde c$ is bounded in the optimization? 2.) Please give timing results for the tuning procedure and the test itself. Otherwise it is impossible to judge the benefit of a linear time test. It seems that one can easily spend a lot of time with parameter tuning. 3.) Why are only very small values of $J$ tested (1 and 5)? It would be interesting to see the scaling for growing $J$, with vs. without optimization. Note that for interpretable results J does not necessarily need to be small, since one can aggregate the weights of features over multiple locations. UPDATE: Thanks to the authors for clarifying all points in the rebuttal.

Confidence in this Review

2-Confident (read it all; understood it all reasonably well)


Reviewer 4

Summary

This paper studies two kernel based test statistics for two-sample testing. Given two i.i.d. samples X and Y of equal size, the problem is to decide whether the population of X is different from Y. Throughout the paper, the authors consider two test named ME test and SCF test, while the theoretical results are provided only for the ME test. ME test is a family of test statistics indexed by two parameters (V, k), and the proposed method chooses these parameters so as to maximize the test power. As an objective for the optimization procedure, a lower bound on the test power is given as a function of the population counterpart of the ME statistic, and the lower bound estimated using the half of the data. This procedure is theoretically justified in the sense of the uniform convergence property of the empirical quantities. The interesting point of the paper is that the optimized parameter V represents the “locations” that distinguish the two probabilities well, which provides a nice interpretability of the testing. Overall, the paper is well structured and well written. The theoretical result, mainly based on the empirical process theory, seems to be correct to the best of my ability to check. The experiments are sufficient for the purpose, and the settings are thoroughly explained.

Qualitative Assessment

The inequality of the Theorem 2 bounds |sup - sup|, which provides the consistency of the supremum of a certain stochastic process. However, I think this statement slightly conflicts with the expression “converge uniformly over all kernels and all distinct test locations” (line 158). It seems that bounding r.h.s. in the first equation in Appendix C.2. might be sufficient. In fact, we are interested in (V, k) that maximizing the empirical lower bound of the test power, and not in the supremum of the process itself.

Confidence in this Review

2-Confident (read it all; understood it all reasonably well)


Reviewer 5

Summary

The paper first establishes that the population counterpart of ME/SCF estimator \lambda_n constitutes a lower bound for the testing power of ME and SCF tests, and proposes optimizing \lambda_n as a strategy to improve testing power for these tests. The optimization is performed with respect to test location V and kernel parameter \sigma. This gives two major advantages: the testing power, according to the experiments, is greatly increased, and the optimized test locations indicate important places where the distributions differ.

Qualitative Assessment

This paper proposes a novel and well performing method to boost the performance of ME and SCF tests. The experiments look really nice. In particular the performance on real dataset is impressive, as it is rare for theoretically guaranteed statistical tests to perform well on high dimensional vision or NLP datasets. The added capability for visualization is also useful and looks reasonable in the experiments. One minor suggestion is, if possible, to simplify Proposition 1 (i.e. by lower bounding with a simpler expression) as it looks too complicated to be intuitive, and elaborate on the conclusion that \lambda_n should be maximized for testing power for finite n as it does not seem obvious from Proposition 1.

Confidence in this Review

2-Confident (read it all; understood it all reasonably well)


Reviewer 6

Summary

This paper presents a novel approach to feature extraction. The extractor is formulated by the theory of hypothesis testing; specifically, testing whether or not two distributions P and Q are the same, based on the comparison of i.i.d. sets drawn from each distribution. By maximizing the power of the test of the difference between P and Q, features which best explain where P and Q differ can be extracted. For example, when comparing documents of different classes, this feature extractor finds words which best explain the differences in the two document classes. The power of this paper's linear-time algorithms is at least as high as a state-of-the-art quadratic-time algorithm's power.

Qualitative Assessment

There is no conclusion or future work section. Although this is not necessarily required, it is generally a good thing to have as a summary of the work that was done and the potential for expansion. I think replacing the equations of the Theorem statements (which I really didn't get much out of -- they are huge equations that take up a lot of space) with a conclusion & future work section would improve the quality of this work. I would also be interested in a more thorough treatment of runtimes. It is noted throughout the paper that MMD is a quadratic-time method, as opposed to ME and SCF which are claimed to be linear. However, ME-full and SCF-full require pre-optimization using a gradient ascent approach to determine the search locations. It's not clear if this pre-optimization is considered when claiming a linear runtime for ME- and SCF-full. I think this is an interesting and novel take on feature extraction (at least I have never seen something like this before). It seems to perform quite well on extracting interpretable features. I could see how something like this could be added to the "toolbox" of a machine learning practitioner.

Confidence in this Review

2-Confident (read it all; understood it all reasonably well)